# Maximin Relative Improvement: Fair Learning as a Bargaining Problem

Jiwoo Han [1]    Moulinath Banerjee [1]    Yuekai Sun [1]

## Abstract

When deploying a single predictor across multiple subpopulations, we propose a fundamentally different approach: interpreting group fairness as a bargaining problem among subpopulations. This game-theoretic perspective reveals that existing robust optimization methods such as minimizing worst-group loss or regret correspond to classical bargaining solutions and embody different fairness principles. We propose relative improvement, the ratio of actual risk reduction to potential reduction from a baseline predictor, which recovers the Kalai–Smorodinsky solution. Unlike absolute-scale methods that may not be comparable when groups have different potential predictability, relative improvement provides axiomatic justification including scale invariance and individual monotonicity. We establish finite-sample convergence guarantees under mild conditions.

## 1. Introduction

Machine learning models now drive consequential decisions in lending, hiring, and healthcare, often replacing human judgment entirely. When deploying a single unified predictor across diverse subpopulations, optimizing for one group often degrades performance on others. How should we balance these competing objectives?

We approach this question through the lens of *cooperative bargaining theory*. When multiple groups must share a single model, the fairness problem becomes one of negotiation: how should groups compromise from their individual optima to reach a mutually acceptable solution? This game-theoretic perspective reveals that existing robust optimization criteria, such as minimizing worst-case group loss or regret (Sagawa et al., 2019; Agarwal & Zhang, 2022) are equivalent to particular notions of fair compromise.

However, these worst-case criteria rely on absolute comparisons of loss or regret across groups, implicitly assuming that a unit of reduction in loss is equally meaningful for all groups. To illustrate, consider predicting income for white and non-white workers in North Dakota using age as the feature in the 2018 American Community Survey (Ding et al., 2021): non-white workers have roughly ten times more potential improvement—the gap between the unconditional mean predictor and group-optimal performance—than white workers. Ensuring equal absolute reductions would extract nearly all available signal from the white group while poorly serving the non-white group.

We address this through a new criterion, *relative improvement*, defined as the fraction of the gap between baseline and optimal performance each model captures. The baseline predictor $f_0$ represents a natural default, such as the mean response in regression or the majority class in classification. Let $f_P^* \in \mathcal{F}$ denote the optimal predictor for distribution $P$:

$$\rho_P(f) = \frac{R_P(f_0) - R_P(f)}{R_P(f_0) - R_P(f_P^*)}, \qquad (1)$$

where $R_P(\cdot)$ is the risk under a given loss function. This criterion applies broadly across regression, classification, and nonparametric settings. We seek models that maximize the worst-case relative improvement:

$$f_{\mathrm{RI}} = \arg \max_{f \in \mathcal{F}} \min_{P \in \mathcal{P}} \rho_P(f), \qquad (2)$$

where $\mathcal{P}$ denotes the set of probability distributions, $\rho^* = \min_{P \in \mathcal{P}} \rho_P(f_{\mathrm{RI}})$ captures equitable signal extraction: every group attains at least this fraction of its achievable risk reduction, independent of task difficulty.

Figure 1 revisits the North Dakota example using this definition. Minimax regret yields $(-74\%, 82\%)$: it allocates nearly all model capacity to the non-white group and leaves the white group worse than baseline. Maximin relative improvement equalizes proportional gains at $(57\%, 57\%)$. This demonstrates that absolute worst-case criteria can yield models extracting vastly unequal fractions of available signal, whereas relative improvement avoids this failure mode.

We establish that the maximin relative improvement criterion corresponds exactly to the *Kalai-Smorodinsky bargaining solution* (Kalai & Smorodinsky, 1975), inheriting its

[1]Department of Statistics, University of Michigan, Ann Arbor, USA. Correspondence to: Jiwoo Han <jiwoohan@umich.edu>.

*Proceedings of the 43rd International Conference on Machine Learning*, Seoul, South Korea. PMLR 306, 2026. Copyright 2026 by the author(s).

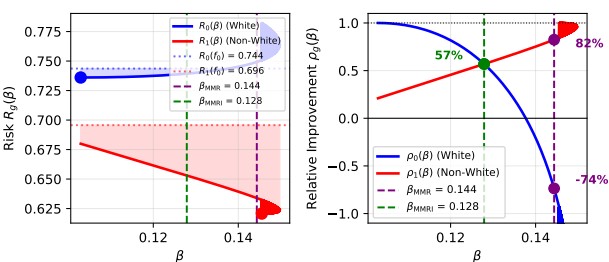

*Figure 1.* Left: per-group risk $R_g(\beta)$. Right: relative improvement $\rho_g(\beta)$, where minimax regret yields $(-74\%, 82\%)$ while maximin relative improvement achieves $(57\%, 57\%)$.

axiomatic properties: Pareto optimality, symmetry, scale invariance, and individual monotonicity. Section 3 develops this connection formally and situates existing robust optimization approaches within this unified bargaining framework, enabling axiomatic comparison of different fairness criteria through their game-theoretic interpretations in Section 4.

Our main contributions are as follows:

1. **Relative improvement as a fairness criterion.** When groups differ in inherent predictability, absolute criteria are not comparable across groups. We propose relative improvement $\rho_g(f)$ and show that maximizing its worst case is exactly the Kalai–Smorodinsky bargaining solution, providing an axiomatic justification.

2. **A unified bargaining framework.** We show that existing robust optimization methods correspond to classical bargaining solutions under a common mapping. Theorem 3.4 establishes the geometric well-posedness needed to import bargaining theory into fair learning.

3. **Axiomatic characterization as a decision guide.** Each criterion is uniquely characterized by a distinct subset of axioms (Table 1), making their normative commitments explicit and directly comparable within a single framework.

4. **Learning-specific guarantees.** We prove that relative improvement satisfies individual rationality (no group is harmed relative to baseline), unlike minimax regret, and establish an $O(1/\sqrt{n})$ finite-sample convergence guarantee.

## 2. Problem Formulation

We consider a general prediction setting with $m$ groups indexed by $g \in \mathcal{G} = \{1, \dots, m\}$. Each group has distribution $P_g$ over $\mathcal{X} \times \mathcal{Y}$, where $X \in \mathcal{X}$ denotes covariates and $Y \in \mathcal{Y}$ is the response variable. Our goal is to find $f \in \mathcal{F}$

using $X$ to predict $Y$ while extracting the most signal in the worst-group case to ensure fairness.

Given a loss function $\ell(y, f(x))$, the group risk is $R_g(f) = \mathbf{E}_{P_g}[\ell(Y, f(X))]$. Let $f_0$ denote a baseline predictor and $f_g^*$ the group-optimal predictor within a function class $\mathcal{F}$.[1] We assume $R_g(f_0) > R_g(f_g^*)$ for all $g \in \mathcal{G}$, ensuring nontrivial prediction problems. The optimization with finite groups is:

$$
\begin{aligned}
f_{\text{RI}} &= \arg\max_{f \in \mathcal{F}} \min_{g \in \mathcal{G}} \rho_g(f) \\
&= \arg\max_{f \in \mathcal{F}} \min_{g \in \mathcal{G}} \frac{R_g(f_0) - R_g(f)}{R_g(f_0) - R_g(f_g^*)},
\end{aligned}
\tag{3}
$$

where $\rho_g(f)$ denotes the relative improvement for group $g$. The loss function and baseline choice depend on the application. We present three representative instantiations below, illustrating the framework's versatility across learning tasks and optimization contexts.

**Parametric Linear Regression.** Consider $\mathcal{F} = \{f(x) = \theta^\top x : \theta \in \Theta\}$ with convex compact $\Theta$ and squared loss. Assume that $Y = \beta_g^\top X + \epsilon_g$ with $\mathbf{E}_g[\epsilon_g | X] = 0$, $\epsilon_g \sim \mathcal{N}(0, \sigma_g^2)$, $\mathbf{E}_g[X] = 0$ and $\mathbf{E}_g[X X^\top] = \Sigma_g$. The baseline is $f_0(x) = 0$ (the unconditional mean) and the group optimum is $f_g^*(x) = \beta_g^\top x$. Then the optimization reduces to $\theta_{\text{RI}} = \arg\max_{\theta \in \Theta} \min_{g \in \mathcal{G}} (1 - \|\theta - \beta_g\|_{\Sigma_g}^2 / \|\beta_g\|_{\Sigma_g}^2)$.

**Binary Classification.** Consider $\mathcal{F} = \{f(x) = \sigma(\theta^\top x) : \theta \in \Theta\}$ with logistic loss. Assume $Y = \mathbf{1}\{\beta_g^\top X + \epsilon > 0\}$, where $X \sim N(0, \Sigma_g)$ and $\epsilon \sim N(0, \sigma_g^2)$. The natural baseline is $f_0(x) = \pi_0$ where $\pi_0 = P(Y = 1)$ is the overall marginal probability, and the group optimum is $f_g^*(x) = \sigma(\beta_g^\top x)$. The relative improvement criterion applies directly.

**Nonparametric RKHS Setting.** For a positive semidefinite kernel $k : \mathcal{X} \times \mathcal{X} \to \mathbb{R}$ inducing RKHS $\mathcal{H}$, consider $\mathcal{F} = \{f \in \mathcal{H} : \|f\|_{\mathcal{H}} \le R\}$ where $R > 0$ controls function complexity. Define the irreducible error $\sigma_g^2 = \min_{f \in \mathcal{F}} \mathbf{E}_g[(Y - f(X))^2]$. The population-level maximin relative improvement objective applies directly with this definition of group-optimal risk. The RKHS structure enables efficient empirical estimation via kernel regularization, with the representer theorem ensuring finite expansions $\hat{f}(x) = \sum_{i=1}^n \alpha_i k(x_i, x)$.

*Remark* 2.1 (Beyond Fairness). The relative improvement framework naturally addresses any multi-objective problem with competing criteria. When $g \in \mathcal{G}$ represent different evaluation metrics rather than groups, the criterion balances performance across all objectives. While we focus on fairness for concreteness, the framework's scope extends to any setting requiring equitable performance across multiple objectives.

---

[1]See Appendix C for discussion of the baseline predictor $f_0$.

## 2.1. Related Work

**Fairness in Machine Learning.** Ensuring equitable model performance across demographic groups has been widely studied through constrained optimization. Hardt et al. (2016) introduced equalized odds and demographic parity constraints, which Agarwal et al. (2018) extended to cost-sensitive classification problems. In the regression setting, Chzhen et al. (2020) propose a plug-in approach for fair regression under demographic parity. Maity et al. (2021) study whether enforcing fairness constraints helps mitigate biases under subpopulation shift, analyzing fairness as linear constraints on risk profiles. These methods optimize average performance subject to fairness metric constraints. Our work takes a complementary objective-based approach, directly maximizing the worst-group relative improvement rather than imposing constraints.

**Robust Optimization for Group Fairness.** Recent work addresses fairness through worst-case optimization. Group Distributional Robust Optimization (Group DRO) (Sagawa et al., 2019) minimizes the worst-group risk, while minimax group regret (Agarwal & Zhang, 2022) compares each group's gap between its risk and its group-optimal risk in absolute units. In the linear regression setting, Meinshausen & Bühlmann (2015) propose maximizing worst-group explained variance from baseline. In contrast, we define fairness in terms of relative improvement, which accounts for both baseline performance and group-specific optimum in a proportional manner. We unify these robust optimization approaches through a cooperative bargaining framework.

**Game-Theoretic Approaches to Fairness.** A closely related line of work formulates group fairness as a multi-objective optimization problem. Minimax Pareto fairness (Martinez et al., 2020) identifies classifiers that minimize the maximum group risk on the Pareto frontier, while Liang et al. (2021) characterize the fairness-accuracy frontier for a given set of inputs to the algorithm. Other works adopt an adversarial formulation, modeling the minimax problem as a two-player zero-sum game between a learner and an adversary (Diana et al., 2021). Earlier work also drew inspiration from bargaining to motivate preference-based fairness notions (Zafar et al., 2017), though without formulating fair learning itself as a bargaining problem. In contrast, while our framework also adopts a multi-objective view, we interpret fairness through a cooperative bargaining lens, where groups are viewed as players negotiating over performance gains, and explicitly identify fairness criteria as classical bargaining solution concepts.

# 3. Bargaining Problem Perspective

We can view the optimization problem (3) as one in which each group makes concessions to reach a compromise ($f_{\text{RI}}$),

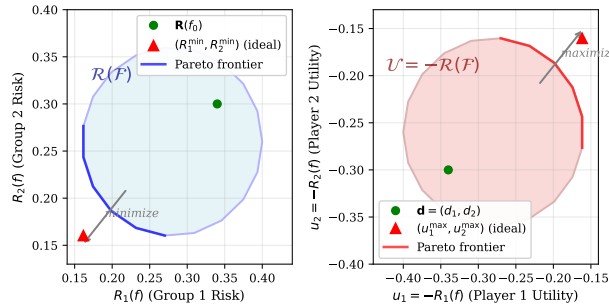

*Figure 2.* Transformation from risk space to utility space. Left: risk space $\mathcal{R}(\mathcal{F})$ where groups aim to minimize risks. Right: utility space $\mathcal{U} = -\mathcal{R}(\mathcal{F})$ where players aim to maximize utilities.

sacrificing their performance from their own optimum ($f_g^*$). This naturally translates into a bargaining problem among $m$ players, where the players are the groups themselves.

## 3.1. Fairness as Bargaining: A General Framework

Any group fairness optimization can be viewed as a bargaining problem among groups. Consider the general form of group fairness optimization:

$$f^* = \arg\max_{f \in \mathcal{F}} \Phi(R_1(f), \ldots, R_m(f)), \qquad (4)$$

where $\Phi : \mathbb{R}^m \to \mathbb{R}$ is an aggregation function determining how to balance group performances. Since $R_g(f)$ represents risk, $\Phi$ must be monotonically decreasing in each argument.

Each predictor $f \in \mathcal{F}$ induces a risk vector of group-specific performances, which we interpret as the outcome of a bargaining game among the $m$ groups, represented by $\boldsymbol{R}(f) = (R_1(f), \ldots, R_m(f))^\top \in \mathbb{R}^m$. The set of all achievable risk profiles, determined by the predictor class $\mathcal{F}$, forms the feasible risk set $\mathcal{R}(\mathcal{F}) = \{\boldsymbol{R}(f) \in \mathbb{R}^m : f \in \mathcal{F}\}$.

To determine which risk profiles represent acceptable compromises, we need a notion of efficiency for comparing vectors. In the bargaining interpretation, an efficient outcome should not allow one group to improve without imposing additional loss on another.

Formally, a risk vector $\boldsymbol{r}'$ Pareto dominates $\boldsymbol{r}$ if $r'_g \leq r_g$ for all groups $g$ with strict inequality for at least one group. A risk vector $\boldsymbol{r}$ is Pareto optimal if no other $\boldsymbol{r}'$ Pareto dominates it. A risk vector $\boldsymbol{r}$ is weakly Pareto optimal if no alternative strictly dominates in all components simultaneously—that is, no $\boldsymbol{r}'$ with $r'_g < r_g$ for all $g$. The collection of all Pareto optimal points forms the Pareto Frontier, the efficiency boundary where no Pareto improvements are possible. Formal definitions are provided in Appendix A.1.

This optimization translates naturally into a bargaining prob-

lem through the following mapping (Figure 2):

$$\text{Groups } g \in \mathcal{G} \leftrightarrow \text{Players}, \tag{5}$$

$$u_g = -R_g(f) \leftrightarrow \text{Utility}, \tag{6}$$

$$d_g = -R_g(f_0) \leftrightarrow \text{Disagreement point}, \tag{7}$$

$$u_g^{\max} = -R_g(f_g^*) \leftrightarrow \text{Ideal point}, \tag{8}$$

$$\mathcal{U} = -\mathcal{R}(\mathcal{F}) \leftrightarrow \text{Feasible set}. \tag{9}$$

Here, $f_0$ represents the baseline predictor, which serves as the disagreement point, i.e., the outcome if groups fail to cooperate and default to the baseline. The ideal point $u_g^{\max} = -R_g(f_g^*)$ represents each group's best achievable utility when optimizing solely for itself.

The key insight is that different fairness approaches implicitly choose different bargaining solutions by specifying how groups negotiate their compromise. Our relative improvement approach, as we show in Section 3.2, corresponds to the Kalai-Smorodinsky solution. In Section 3.3, we show that other robust optimization approaches such as group DRO, maximin relative explained variance, and minimax regret can also be translated into bargaining solutions.

To complete this perspective, it remains to verify that the learning problem induces a well-posed bargaining problem. Classical bargaining solutions are defined over feasible sets that are compact and convex, ensuring existence and axiomatic characterizations of the resulting agreements. In general, an arbitrary function class of predictors does not guarantee these properties. In Section 3.4, we identify mild conditions under which the feasible risk set $\mathcal{R}(\mathcal{F})$ is compact and convex, thereby formally connecting the learning problem and bargaining problem and enabling further analysis of their properties in Section 4.

### 3.2. The Kalai-Smorodinsky Solution and Relative Improvement

Our optimization in Equation (3) corresponds exactly to the Kalai–Smorodinsky (KS) bargaining solution (Kalai & Smorodinsky, 1975), a classical result in cooperative game theory characterized by compelling fairness axioms. This connection grounds our approach in established theory rather than ad-hoc construction.

In the 2-player setting with compact and convex feasible set $\mathcal{U} \subset \mathbb{R}^2$, the KS solution is the unique solution satisfying four axioms—Pareto optimality, symmetry, scale invariance, and individual monotonicity—and selects a Pareto-optimal point $(u_1, u_2)$ equalizing relative improvements:

$$\frac{u_1 - d_1}{u_1^{\max} - d_1} = \frac{u_2 - d_2}{u_2^{\max} - d_2} = \rho^*, \tag{10}$$

where $\rho^*$ is the largest jointly realizable fraction of improvement. Kalai & Smorodinsky (1975) establishes unique exis-

tence of this point. Theorem B.1 adapts this to our statistical setting and shows it coincides with the maximin solution.

For $m > 2$ players (Moulin, 1984), the KS solution generalizes to the maximin solution:

$$\rho^* = \max_{u \in \mathcal{U}} \min_{g \in [m]} \frac{u_g - d_g}{u_g^{\max} - d_g}. \tag{11}$$

When $m > 2$, this may yield multiple or Pareto-dominated solutions (Moulin, 1984). The lexicographic maximin (leximin) extension (Imai, 1983) refines this by imposing a priority structure: among all solutions maximizing the worst group's relative improvement, it selects those maximizing the second-worst group's improvement, and continues sequentially.

*Remark* 3.1 (Uniqueness of risk vectors vs. predictors). The KS solution guarantees a unique risk vector $\boldsymbol{R}(f_{\text{RI}})$, but not necessarily a unique predictor. Since our optimization depends only on group-wise risks, multiple predictors solving the maximin relative improvement problem might achieve the same risk profile, for example in overparameterized settings. Uniqueness of optimal predictor requires additional assumptions such as strict convexity of the loss (see Theorem B.3).

### 3.3. Comparison with Alternative Fairness Approaches

Our bargaining framework is not limited to relative improvement—it provides a unified lens for understanding existing robust fairness methods. We now show how group DRO, maximin explained variance, and minimax regret can also be expressed as bargaining solutions, each corresponding to different fairness principles.

Group distributionally robust optimization (GDRO, Sagawa et al. (2019)) minimizes worst-group risk, maximin explained variance (MMV, Meinshausen & Bühlmann (2015)) maximizes lowest-group improvement from baseline, and minimax regret (MMR, Agarwal & Zhang (2022)) minimizes worst-group regret (Figure 3):

$$f_{\text{GDRO}} = \arg\min_{f \in \mathcal{F}} \max_{g \in \mathcal{G}} R_g(f), \tag{12}$$

$$f_{\text{MMV}} = \arg\max_{f \in \mathcal{F}} \min_{g \in \mathcal{G}} [R_g(f_0) - R_g(f)], \tag{13}$$

$$f_{\text{MMR}} = \arg\min_{f \in \mathcal{F}} \max_{g \in \mathcal{G}} [R_g(f) - R_g(f_g^*)]. \tag{14}$$

Under the mapping in Section 3.1, these correspond to:

GDRO (Rawlsian): $u_{\text{RAW}} = \arg\max_{\boldsymbol{u} \in \mathcal{U}} \min_g u_g$,

MMV (Egalitarian): $u_{\text{EG}} = \arg\max_{\boldsymbol{u} \in \mathcal{U}} \min_g (u_g - d_g)$,

MMR (Equal Loss): $u_{\text{EL}} = \arg\min_{\boldsymbol{u} \in \mathcal{U}} \max_g (u_g^{\max} - u_g)$,

where $\boldsymbol{u} = (u_1, \cdots, u_m)^\top$ denotes the utility vector.

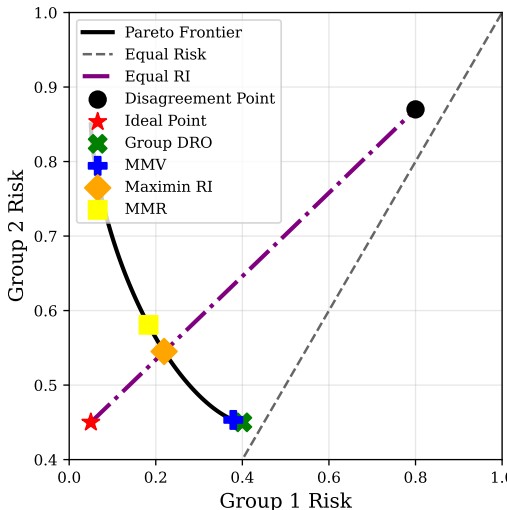

*Figure 3.* Schematic diagram comparing group fairness methods in risk space, after assuming the Pareto Frontier. Each method selects a different solution on the Pareto frontier. Detailed characterization are provided in Appendix F.2.

The Ralwsian solution prioritizes the worst-off group, while the Egalitarian solution maximizes the minimum gain from the disagreement point $d$.[2] Equal loss minimizes maximum regret from the ideal outcomes (Chun, 1988), corresponding to $p = \infty$ case of the general Yu solution which minimizes $(\sum_g |u_g^{\max} - u_g|^p)^{1/p}$ (Yu, 1973). Among classical bargaining solutions, the Nash solution (Nash et al., 1950) is perhaps the most well-known, defined as $u_{\text{Nash}} = \arg\max_{u \in \mathcal{U}} \prod_g (u_g - d_g)$. It maximizes the product of utilities gained from the disagreement point. A comprehensive comparison of axiomatic properties satisfied by each solution and their implications for fairness is provided in Section 4.

### 3.4. Risk Set Properties and Well-Posedness

The bargaining framework established above requires the feasible set to be compact and convex to ensure well-defined solutions. We now provide conditions under which the risk set $\mathcal{R}(\mathcal{F})$ satisfies these properties, enabling us to apply game-theoretic results and derive further properties in Section 4.

**Assumption 3.2** (Parametric setting). Assume the function class can be parametrized as $\mathcal{F} = \{f_\theta : \theta \in \Theta\}$ where (i) the loss function $\ell(y, f_\theta(x))$ is convex and continuous with respect to $\theta$, (ii) there exists an integrable function $h : \mathcal{X} \times \mathcal{Y} \to \mathbb{R}_+$ such that $0 \le \ell(y, f_\theta(x)) \le h(x, y)$ for all $\theta \in \Theta$ and $\mathbf{E}_g[h(X, Y)] < \infty$ for all $g \in \mathcal{G}$, and (iii) the

<hr>

[2]The egalitarian principle seeks to equalize utilities, but when full equalization is infeasible or Pareto-inefficient, it selects the most egalitarian feasible distribution by lexicographically comparing utility profiles in increasing order (Moulin, 2004, Sec. 3.3).

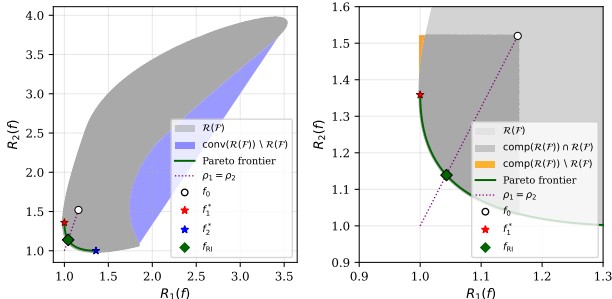

*Figure 4.* Risk set $\mathcal{R}(\mathcal{F})$, its convex hull (left), and comprehensive closure (right) for the linear regression setting (in Section 2) with $\Theta = \{\theta : \|\theta\| \le 1\}$, $\beta_1 = (0.4, 0)$, $\beta_2 = (0.4, 0.6)$, $\Sigma_1 = \left(\begin{smallmatrix} 1 & 0.5 \\ 0.5 & 1 \end{smallmatrix}\right)$, $\Sigma_2 = \left(\begin{smallmatrix} 1 & 0 \\ 0 & 1 \end{smallmatrix}\right)$, and $\sigma_g = 1$. Note that $\text{conv}(\mathcal{R}(\mathcal{F})) \setminus \mathcal{R}(\mathcal{F})$ and $\text{comp}(\mathcal{R}(\mathcal{F})) \setminus \mathcal{R}(\mathcal{F})$ contain no Pareto optimal points.

parameter space $\Theta$ is compact and convex.

**Assumption 3.3** (Nonparametric setting). Let $\mathcal{F}$ be a class of real-valued functions on $\mathcal{X}$ where (i) the loss function $\ell(y, t)$ is convex and continuous with respect to $t$, (ii) there exists an integrable function $h : \mathcal{X} \times \mathcal{Y} \to \mathbb{R}_+$ such that $0 \le \ell(y, f(x)) \le h(x, y)$ for all $f \in \mathcal{F}$ and $\mathbf{E}_g[h(X, Y)] < \infty$ for all $g \in \mathcal{G}$, and (iii) the function class $\mathcal{F}$ is convex, and compact under a topology for which evaluation maps $f \mapsto f(x)$ are continuous for every $x \in \mathcal{X}$.

The examples introduced in Section 2 satisfy the assumptions under standard regularity conditions. Detailed verifications are provided in Appendix D. More generally, sufficient conditions for verifying Assumptions 3.2(ii) and 3.3(ii) are provided in Appendix B.1 (Proposition B.4).

**Theorem 3.4** (Properties of the risk set). *Under either Assumption 3.2 or Assumption 3.3, the risk set $\mathcal{R}(\mathcal{F})$ satisfies:*

1. *$\mathcal{R}(\mathcal{F})$ is a compact set.*

2. *Let $\text{conv}(\mathcal{R}(\mathcal{F}))$ denote the convex hull of $\mathcal{R}(\mathcal{F})$. Then, $\text{conv}(\mathcal{R}(\mathcal{F})) \setminus \mathcal{R}(\mathcal{F})$ does not contain any Pareto optimal point.*

*When the loss function is additionally strictly convex (in $\theta$ or $t$, respectively), the following properties also hold:*

3. *Every weakly Pareto optimal point in $\mathcal{R}(\mathcal{F})$ is Pareto optimal.*

4. *The set $\text{conv}(\mathcal{R}(\mathcal{F})) \setminus \mathcal{R}(\mathcal{F})$ does not contain any weakly Pareto optimal point.*

Theorem 3.4 shows that while $\mathcal{R}(\mathcal{F})$ need not be convex, its convex hull $\text{conv}(\mathcal{R}(\mathcal{F}))$ is compact and convex and—crucially—contains no (weakly) Pareto optimal points outside $\mathcal{R}(\mathcal{F})$ (see Figure 4). Thus, convexification constitutes an exact relaxation: although the bargaining problem

is formulated over $\text{conv}(\mathcal{R}(\mathcal{F}))$, all solutions selected by standard bargaining criteria correspond to realizable risks in $\mathcal{R}(\mathcal{F})$ induced by some $f \in \mathcal{F}$. Classical bargaining theory takes compactness and convexity of the feasible set as primitive assumptions; our contribution is to derive these geometric properties directly from the function class and standard regularity conditions on the loss.

Beyond well-posedness, this result serves as a general bridge between cooperative bargaining theory and fair learning: any bargaining solution that selects Pareto optimal outcome can be imported into the fair learning setting whenever Assumptions 3.2 or 3.3 holds, without requiring case-by-case verification of geometric well-posedness. We now turn to the axiomatic properties within this framework.

# 4. Fairness Guarantees of Relative Improvement

Why is relative improvement a principled fairness criterion? It ensures no group is harmed relative to baseline, and it satisfies Pareto optimality, scale invariance, and individual monotonicity that alternatives lack.

## 4.1. No-Harm Guarantee

When the baseline predictor $f_0 \in \mathcal{F}$, the maximin relative improvement solution naturally ensures no group is harmed.

**Theorem 4.1** (Bounded Loss Relative to Baseline). *Assume the baseline predictor $f_0 \in \mathcal{F}$. Then the maximin relative improvement solution $f_{RI}$ from (3) satisfies*

$$R_g(f_{RI}) \leq R_g(f_0), \quad \text{for all } g \in \mathcal{G}. \tag{15}$$

This property corresponds to *individual rationality* in bargaining theory: each group performs at least as well as the disagreement point (baseline).

Notably, minimax regret violates this property. Revisiting the motivating example in Figure 1, when group 1 has less potential improvement than group 2, minimizing worst-group regret can increase the risk of the less signal group, yielding $R_1(\theta_{\text{Regret}}) > R_1(0)$. In contrast, relative improvement's normalization prevents such harm.

This bounded-loss guarantee is related to constraints used in some fair learning methods (Agarwal et al., 2019; Chzhen & Schreuder, 2022), though those approaches typically require explicit bounded-loss constraints as part of their formulation. In contrast, our framework obtains this property directly from the maximin objective without additional constraints.

## 4.2. Axiomatic Characterization: $m = 2$ case

We first characterize axioms of the two group case, which corresponds to a 2-player KS solution in equation (10).

**Proposition 4.2** (Relative improvement as the KS solution ($m = 2$), adapted from Kalai & Smorodinsky (1975)). *For $m = 2$, $f_{RI}$ in Equation (3) satisfies:*

1. ***Pareto Optimality (PO).*** *No $f' \in \mathcal{F}$ satisfies $\rho_g(f') \geq \rho_g(f_{RI})$ for all $g$ with strict inequality for at least one.*

2. ***Symmetry (SYM).*** *For any permutation $\pi : \mathcal{G} \to \mathcal{G}$, denote $\pi * \mathcal{R}(\mathcal{F}) = \{r \in \mathbb{R}^m : r_{\pi(g)} = s_g \text{ for all } g \text{ for some } s \in \mathcal{R}(\mathcal{F})\}$. Under any permutation $\pi$, $\pi * R(f_{RI}) = R(f_{\pi,RI})$, where $f_{\pi,RI}$ denotes a solution with the permuted feasible set $\pi * \mathcal{R}(\mathcal{F})$.*

3. ***Scale Invariance (SI).*** *Affine transformations $\widetilde{R}_g(f) = c_g R_g(f) + a_g$ (with $c_g > 0$) preserve relative improvements: $\widetilde{\rho}_g(\tilde{f}_{RI}) = \rho_g(f_{RI})$.*

4. ***Individual Monotonicity (IM).*** *For all $g$, if $\mathcal{F}_1 \subseteq \mathcal{F}_2$ with the same baseline $f_0$ and optimal risk for group $g' \neq g$, then $R_g(f_{2,RI}) \leq R_g(f_{1,RI})$, ensuring that enlarging the feasible set while keeping other groups' optimum cannot harm any group.*

*The relative improvement metric is the unique solution satisfying (PO), (SYM), (SI), and (IM) for $m = 2$.*

These axioms ensure: efficiency (PO), equal treatment of groups (SYM), measurement-independence (SI), and that more options help every group (IM).

## 4.3. Axiomatic Characterization: $m > 2$ case

For $m > 2$ groups, Roth (1979) showed no solution can satisfy all four axioms (PO, SYM, SI, IM) simultaneously. The leximin refinement of the KS solution (Imai, 1983) resolves this impossibility by modifying the individual monotonicity axiom, while relying on comprehensive feasible sets.

**Leximin refinement.** Let $\rho_{(1)}(u) \leq \cdots \leq \rho_{(m)}(u)$ denote the sorted relative improvements (note that the group ordering may differ across $u$):

$$u^* = \arg\max_{u \in \mathcal{U}}^{\text{lex}} \left( \rho_{(1)}(u), \ldots, \rho_{(m)}(u) \right), \tag{16}$$

where the lexicographic order prioritizes earlier components: $u \succ u'$ if $\rho_{(i)}(u) > \rho_{(i)}(u')$ at the first differing index $i$. This refinement ensures a unique risk vector and Pareto optimality in multi-group settings.

**Comprehensive risk set.** Classical bargaining theory for multi-players assumes a *comprehensive* feasible set: if $u \in \mathcal{U}$ and $d \leq u' \leq u$ componentwise, then $u' \in \mathcal{U}$ (voluntary utility disposal). Our risk sets $\mathcal{R}(\mathcal{F})$ are typically not comprehensive, as risk cannot be voluntarily increased. However, comprehensiveness is not restrictive in our setting, as established by the following result.[3]

---

[3]See Appendix E.2 for further discussion of leximin refinement and comprehensive closures.

**Theorem 4.3** (Equivalence under comprehensive closure)**.** *When $f_0 \in \mathcal{F}$, the leximin solution on $\mathcal{R}(\mathcal{F})$ coincides with that on its comprehensive closure $\text{comp}(\mathcal{R}(\mathcal{F})) = \{\mathbf{r}' : \exists \mathbf{r} \in \mathcal{R}(\mathcal{F}), -d_g \geq r'_g \geq r_g \; \forall g\}$, where $-d_g = R_g(f_0)$ denotes the baseline risk for group $g$.*

**Proposition 4.4** (Relative improvement as the KS solution ($m > 2$), adapted from Imai (1983))**.** *For $m > 2$ and any baseline $f_0 \in \mathcal{F}$, $f_{RI}$, a leximin relative improvement solution, satisfies:*

1. ***Pareto Optimality (PO).*** *(Same as Proposition 4.2.)*

2. ***Symmetry (SYM).*** *(Same as Proposition 4.2.)*

3. ***Scale Invariance (SI).*** *(Same as Proposition 4.2.)*

4. ***Independence of Irrelevant Alternatives with Ideal point (IIIA).*** *If $\text{comp}(\mathcal{R}(\mathcal{F}_1)) \subseteq \text{comp}(\mathcal{R}(\mathcal{F}_2))$ with the same baseline $f_0$ and the same optimal risk profile $(R_g(f_g^*))_{g \in \mathcal{G}}$, and if $\mathbf{R}(f_{2,RI}) \in \text{comp}(\mathcal{R}(\mathcal{F}_1))$, then $\mathbf{R}(f_{1,RI}) = \mathbf{R}(f_{2,RI})$.*

5. ***Modified Individual Monotonicity (IM').*** *Define the projection ${}^g\mathcal{R}(\mathcal{F}) = \{(r_1, \ldots, r_{g-1}, r_{g+1}, \ldots, r_m) : r \in \mathcal{R}(\mathcal{F})\}$. If $\text{comp}(\mathcal{R}(\mathcal{F}_1)) \subseteq \text{comp}(\mathcal{R}(\mathcal{F}_2))$ with the same baseline $f_0$ and $\text{comp}({}^g\mathcal{R}(\mathcal{F}_1)) = \text{comp}({}^g\mathcal{R}(\mathcal{F}_2))$, then $R_g(f_{2,RI}) \leq R_g(f_{1,RI})$.*

*The relative improvement metric is the unique solution satisfying (PO), (SYM), (SI), (IIIA), and (IM') for $m > 2$.*

The condition $\text{comp}(\mathcal{R}(\mathcal{F}_1)) \subseteq \text{comp}(\mathcal{R}(\mathcal{F}_2))$ appearing in axioms (IIIA) and (IM') means that any predictor in $\mathcal{F}_1$ can be weakly Pareto dominated by some predictor in $\mathcal{F}_2$. Thus, (IM') states that Pareto-improving expansions of the feasible set that leave other groups unchanged should not worsen group $g$'s relative improvement.

In summary, extending to $m > 2$ groups requires two refinements. The leximin criterion resolves non-existence and non-uniqueness by hierarchically prioritizing relative improvements. Comprehensive closure ensures compatibility with classical bargaining axioms without changing the solution. Together, these yield a unique, scale-invariant, and stable fairness criterion.

### 4.4. Comparison with Alternative Bargaining Solutions

To understand why the axioms satisfied by relative improvement are desirable, we compare them with alternative bargaining solutions introduced in Section 3.3. Each solution satisfies different axioms (detailed definitions in Appendix E.1).

Table 1 compares the axiomatic properties of bargaining-based fairness solutions across various numbers of groups.

Since absolute-scale approaches like Group DRO, minimax regret, and maximin explained variance share the same axiomatic structure but differ only in their reference points ($\boldsymbol{d}$ or $\boldsymbol{u}^*$), the table reports explained variance and regret as representative cases. For $m > 2$, the axiomatic characterization of absolute-scale methods are formulated for normalized bargaining problems; after rescaling by potential improvement, the resulting problem is uniquely solved by the KS solution (Chen, 2000).

**Why Full Pareto Optimality Matters.** GDRO, MMV, and MMR only guarantee weak PO, potentially selecting dominated solutions. Full PO ensures no missed opportunities to improve some groups without harming others.

**Why Individual Monotonicity Matters.** Nash solution violates IM: more expressive models may harm some groups. In contrast, IM guarantees that richer function classes weakly benefit all groups in risks.

As discussed in Section 1, scale invariance is essential when groups have different baseline predictability. Only KS satisfies (PO), (SYM), (SI), and (IM) simultaneously—a compelling combination for fair learning.

*Remark* 4.5 (Axiom-based criterion selection)*.* No single fairness criterion dominates all others across every setting. Indeed, most criteria in Table 1 select Pareto-optimal risk profiles, so they cannot be ranked by Pareto dominance alone; the table instead shows that each is uniquely characterized by a distinct subset of axioms. Rather than asserting that relative improvement is universally preferable, we view the axiomatic framework as a *decision guide*: a practitioner should first identify which axioms are most relevant to their context, and the solution is then uniquely determined. For example, when groups differ substantially in inherent predictability, scale invariance is essential—making relative improvement the natural choice. Conversely, a practitioner who prioritizes strong monotonicity over scale invariance is led to the egalitarian solution instead.

## 5. Empirical Estimator

We now turn to finite-sample estimation and its empirical counterparts. Given $n$ samples where group $g$ has $n_g$ observations $\{(x_{ig}, y_{ig})\}_{i=1}^{n_g}$, the empirical risk for group $g$ is $\hat{R}_g(f) = 1/n_g \sum_{i=1}^{n_g} \ell(y_{ig}, f(x_{ig}))$.

Let $f_0$ denote a baseline predictor. The empirical group-optimal predictor is $\hat{f}_g^* \in \arg\min_{f \in \mathcal{F}} \hat{R}_g(f)$ with risk $\hat{R}_g^* = \hat{R}_g(\hat{f}_g^*)$.

For each group $g$, the empirical relative improvement is

$$\hat{\rho}_g(f) = \frac{\hat{R}_g(f_0) - \hat{R}_g(f)}{\hat{R}_g(f_0) - \hat{R}_g^*}, \tag{17}$$

and we define an empirical maximin predictor as $\hat{f}_{\text{RI}} \in$

*Table 1.* Axiomatic comparison of bargaining-based fairness solutions.

| SOLUTION | PO | SYM | SI | IIA | IM | TI | SM |
|---|---|---|---|---|---|---|---|
| *Relative Improvement (KS-based)* | | | | | | | |
| $m = 2$, MAXIMIN = EQUAL (KALAI & SMORODINSKY, 1975) | √ | √ | √ | × | √ | ○ | × |
| $m > 2$, LEXIMIN (IMAI, 1983) | √ | √ | √ | □ | □ | ○ | × |
| *Nash Bargaining Solution* | | | | | | | |
| GENERAL $m$ (NASH ET AL., 1950) | √ | √ | √ | √ | × | ○ | × |
| *Explained Variance (Egalitarian-based)* | | | | | | | |
| EQUAL (KALAI, 1977) | △ | √ | × | × | ○ | √ | √ |
| $m = 2$, MAXIMIN IF EQUAL = MAXIMIN (CHEN, 2000) | √ | √ | × | × | ○ | √ | √ |
| *Regret (Equal Loss-Based)* | | | | | | | |
| EQUAL (CHUN, 1988) | △ | √ | × | × | × | √ | □ |

**Axioms:** PO = Pareto Optimality; SYM = Symmetry; SI = Scale Invariance; IIA = Independence of Irrelevant Alternatives; IM = Individual Monotonicity; TI = Translation Invariance; SM = Strong Monotonicity.
√: satisfied and used for characterization; □: satisfied after modification and used for characterization; ○: satisfied but not used; △: weak Pareto optimality; ×: violated.

$\arg\max_{f \in \mathcal{F}} \min_{g \in \mathcal{G}} \hat{\rho}_g(f)$. The population relative improvement value using such empirical optimal predictor is

$$\rho_g(\hat{f}_{\mathrm{RI}}) = \frac{R_g(f_0) - R_g(\hat{f}_{\mathrm{RI}})}{R_g(f_0) - R_g(f_g^*)}. \quad (18)$$

To establish convergence guarantees for $\hat{f}_{\mathrm{RI}}$, we require a uniform concentration of empirical risks around their population counterparts.

**Assumption 5.1** (Uniform concentration for group risks). For each group $g \in \mathcal{G}$ and all $t > 0$, there exists a nondecreasing function $r_{n_g}(t)$ such that, with probability at least $1 - 2e^{-t}$,

$$\sup_{f \in \mathcal{F}} \left| \hat{R}_g(f) - R_g(f) \right| \leq r_{n_g}(t).$$

The following lemma provides concrete sufficient conditions under which Assumption 5.1 holds.

**Lemma 5.2** (Sufficient conditions for Assumption 5.1). *Assumption 5.1 holds under the following conditions:*

- *(Lipschitz Loss) $\ell(y, z)$ is $L$-Lipschitz in $z$;*

- *(Entropy Condition) There exist $C_0 > 0$ and $p \in [0, 2)$ with $\log \mathcal{N}(\varepsilon, \mathcal{F}, L_2(P_{g,X})) \leq C_0\, \varepsilon^{-p}$ for every $g \in \mathcal{G}$, where $P_{g,X}$ denotes the marginal distribution of $P_g$ on $X$.*

- *Either (A) (Bounded Loss) $\ell$ is bounded in $[0, 1]$, or (B) (Sub-Gaussian Envelope) $\sup_{f \in \mathcal{F}} |\ell(y, f(x))|$ is sub-Gaussian with parameter $\sigma > 0$ under each $P_g$.*

*Then for some constants $C_1, C_2 > 0$,*

$$r_{n_g}(t) \leq C_1 \frac{L}{\sqrt{n_g}} + C_2 \sqrt{\frac{t}{n_g}}.$$

We also require that the baseline predictor is improvable for each group.

**Assumption 5.3** (Positive improvement bound). For every group $g \in \mathcal{G}$, there exists potential for improvement: $R_g(f_0) - R_g(f_g^*) > 0$ where $f_g^* \in \arg\min_{f \in \mathcal{F}} R_g(f)$. Define $\Delta := \min_{g \in \mathcal{G}} \left( R_g(f_0) - R_g(f_g^*) \right) > 0$.

With these assumptions in place, we can now state our main convergence result.

**Theorem 5.4** (Convergence Rate of Relative Improvement). *Let*

$$\hat{f}_{RI} \in \arg\max_{f \in \mathcal{F}} \min_{g \in \mathcal{G}} \hat{\rho}_g(f), \quad f_{RI} \in \arg\max_{f \in \mathcal{F}} \min_{g \in \mathcal{G}} \rho_g(f).$$

*Under Assumptions 5.1 and 5.3, for any $\delta \in (0, 1)$, with probability at least $1 - \delta$,*

$$\min_{g \in \mathcal{G}} \rho_g(\hat{f}_{\mathrm{RI}}) \geq \min_{g \in \mathcal{G}} \rho_g(f_{\mathrm{RI}}) - \frac{C}{\Delta} \max_{g \in \mathcal{G}} r_{n_g}\big( \log(2m/\delta) \big) \quad (19)$$

*for some constant $C > 0$. We assume $n_g$ is sufficiently large that $\max_{g \in \mathcal{G}} r_{n_g}\big( \log(2m/\delta) \big) < \Delta/4$ holds.*

The convergence guarantee in Theorem 5.4 depends on the uniform concentration rate $r_{n_g}(t)$, which by Lemma 5.2 scales as $O(1/\sqrt{n_g})$ under standard regularity conditions. This rate matches those established for alternative fairness criteria such as minimax regret (Agarwal & Zhang, 2022; Mo et al., 2024).

## 6. Experiment

We use the 2018 American Community Survey (ACS) accessed via `folktables` (Ding et al., 2021), predicting log-transformed personal income for full-time employees across all 50 U.S. states. We consider two binary group

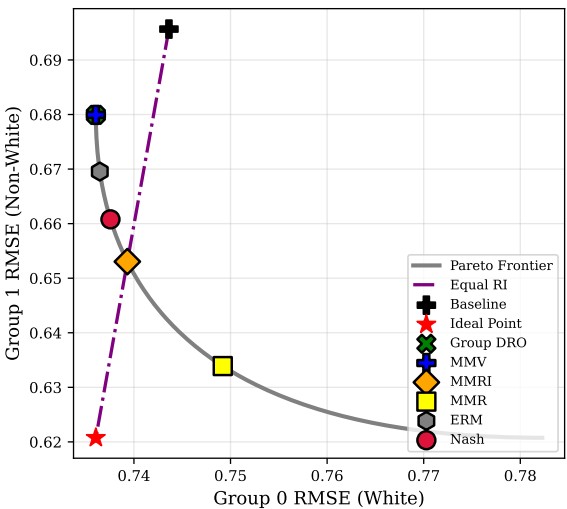

*Figure 5.* **North Dakota, race partition, age (`AGEP`), gap ratio ≈ 0.10.** Age is roughly ten times more predictive for non-white workers than for white workers.

partitions (sex, race) and four features (age, education, marital status, householder status), yielding 400 configurations in total. For each, we fit group-reweighted linear models by sweeping $\lambda \in [0, 1]$ over 10,001 values to trace the full Pareto frontier; see Appendix G for details.

**Differential predictability across groups is common in practice.** For each configuration, we compute the oracle gap ratio $r = (R_0(f_0) - R_0(f_0^*))/(R_1(f_0) - R_1(f_1^*))$, measuring the relative possible improvement of the two groups. Across all 400 configurations, the gap ratio ranges from 0.04 to 3.48, with a median of 0.80. Only 27% of configurations are near-symmetric ($0.8 \leq r \leq 1.25$); the remaining 73% exhibit moderate-to-extreme asymmetry, with 26% exceeding a factor of two ($r < 0.5$ or $r > 2.0$). Asymmetry is especially pronounced under the race partition, where features such as marital status and householder status yield extreme ratios in 50% and 86% of states, respectively. These results confirm that differential predictability across groups is a pervasive feature of real data, not an artifact of synthetic construction.

**Consequences for fairness criteria.** Figure 5 shows the Pareto frontier and method solutions for the North Dakota race partition with age(`AGEP`) as the single feature (gap ratio ≈ 0.10). All absolute-scale methods (MMR, Group DRO, MMV) deviate substantially from the equal-RI line, with MMR leaving the white group worse than baseline. MMRI lies on the equal-RI line by construction. Full results across states, features, and the four-feature model are provided in Appendix G.

## 7. Discussion

This work views group fairness through a cooperative bargaining lens, complementing worst-case optimization approaches. By interpreting performance trade-offs as negotiated compromises rather than worst-case competition, we connect fairness criteria to classical bargaining solutions. Applying bargaining theory to fairness optimization requires verifying that feasible risk sets satisfy the geometric assumptions of cooperative game theory. We provide sufficient conditions for convexity and compactness (Theorem 3.4), covering parametric and nonparametric function classes, thereby offering a unified framework for applying bargaining solutions to diverse learning problems.

Relative improvement inherits the axiomatic properties of the Kalai–Smorodinsky solution, providing principled justification without ad-hoc penalty choices. In the two-group case, it recovers the maximin principle while yielding a unique Pareto-optimal compromise, and preserving scale invariance and individual monotonicity that regret-based methods lack due to their absolute-scale formulation. When a feasible baseline predictor is available, the solution ensures no group is harmed, a guarantee minimax regret violates. Moreover, under standard assumptions, our empirical estimator achieves $O(1/\sqrt{n})$ convergence rates, matching those of regret-based methods. In practice, relative improvement is particularly suited to settings with heterogeneous group predictability, where absolute regret fails to account for differences in task difficulty.

This work establishes a broad theoretical framework for relative improvement. While prior fairness criteria have been studied within specific models, such instantiations of our framework remain unexplored. Additionally, while we propose leximin extensions for multiple groups, we do not address the algorithmic challenges of computing these solutions efficiently, which grows in complexity with the number of groups. These refinements are left for future investigation, as our primary contribution lies in introducing and formalizing the metric itself.

Several more general directions remain open. Extending relative improvement to overparameterized regimes raises fundamental difficulties shared by robust optimization methods: strict convexity may fail, geometric properties weaken, and standard complexity controls such as covering numbers become inapplicable, precluding uniform convergence guarantees. Understanding fairness and robustness under this regime is an important direction for future work. Finally, the bargaining viewpoint naturally extends beyond fairness to general multi-objective optimization, suggesting promising applications in settings where objectives are heterogeneous and not directly comparable—for instance, in large language model evaluation across diverse benchmarks.

## Impact Statement

This paper presents work whose goal is to advance the field of Machine Learning. There are many potential societal consequences of our work, none which we feel must be specifically highlighted here.

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

# A. Definitions

## A.1. Pareto Optimality

We provide formal definitions of Pareto optimality concepts used in Section 3.

**Definition A.1** (Risk vector). Each predictor $f \in \mathcal{F}$ induces a risk vector $\boldsymbol{R}(f) = (r_1, \ldots, r_m)^\top \in \mathbb{R}^m$, where $r_g = R_g(f)$ denotes the risk incurred by group $g$.

**Definition A.2** (Pareto dominance). For risk vectors $\boldsymbol{r}, \boldsymbol{r}' \in \mathbb{R}^m$, we say $\boldsymbol{r}'$ Pareto dominates $\boldsymbol{r}$ (written $\boldsymbol{r}' \prec \boldsymbol{r}$) if $r'_g \leq r_g$ for all $g \in \mathcal{G}$ with strict inequality for at least one group.

**Definition A.3** (Weak Pareto dominance). For risk vectors $\boldsymbol{r}, \boldsymbol{r}' \in \mathbb{R}^m$, we say $\boldsymbol{r}'$ weakly Pareto dominates $\boldsymbol{r}$ if $r'_g < r_g$ for all $g \in \mathcal{G}$.

**Definition A.4** (Pareto optimality). Let $S \subseteq \mathbb{R}^m$. A point $\boldsymbol{r} \in S$ is Pareto optimal in $S$ if there exists no $\boldsymbol{r}' \in S$ such that $\boldsymbol{r}' \prec \boldsymbol{r}$. Equivalently, $\boldsymbol{r}$ is Pareto optimal if any improvement for one group requires degradation for another.

**Definition A.5** (Weak Pareto optimality). Let $S \subseteq \mathbb{R}^m$. A point $\boldsymbol{r} \in S$ is weakly Pareto optimal in $S$ if there exists no $\boldsymbol{r}' \in S$ such that $r'_g < r_g$ for all $g \in \mathcal{G}$. Equivalently, $\boldsymbol{r}$ is weakly Pareto optimal if no alternative in $S$ strictly dominates it in all components simultaneously.

**Definition A.6** (Feasible risk set). The feasible risk set is $\mathcal{R}(\mathcal{F}) = \{\boldsymbol{R}(f) : f \in \mathcal{F}\}$.

**Definition A.7** (Pareto frontier of risk set). The Pareto frontier of the risk set $\mathcal{R}(\mathcal{F})$ is $\mathcal{P}_\mathcal{R}(\mathcal{F}) = \{\boldsymbol{r} \in \mathcal{R}(\mathcal{F}) : \boldsymbol{r}$ is Pareto optimal in $\mathcal{R}(\mathcal{F})\}$. The collection of all Pareto optimal points forms the Pareto frontier, i.e., the efficiency boundary where no Pareto improvements are possible.

## A.2. Risk to Relative Improvement

We provide formal definitions of the risk to relative improvement transformation and associated concepts used in the proof of Theorem B.1.

**Definition A.8** (Risk-to-Relative-Improvement Transformation). Let $\boldsymbol{r} = (r_1, \ldots, r_m)^T \in \mathbb{R}^m$ be a risk vector. The transformation from risk space to relative improvement space is defined as

$$T : \mathbb{R}^m \to \mathbb{R}^m, \quad T(\boldsymbol{r}) = \boldsymbol{\rho}$$

where

$$\rho_g = \frac{r_g^0 - r_g}{r_g^0 - r_g^*}$$

for each $g \in \mathcal{G}$. Here $r_g^*$ denotes the group-optimal risk (the minimum risk achievable for group $g$ over $\mathcal{F}$) and $r_g^0$ denotes the baseline risk. We assume $r_g^* < r_g^0$ for all $g$ to ensure the transformation is well-defined.

**Definition A.9** (Relative improvement vector). Given a predictor $f \in \mathcal{F}$ with risk vector $\boldsymbol{R}(f)$, its relative improvement vector is defined as $T(\boldsymbol{R}(f)) \in \mathbb{R}^m$.

**Definition A.10** (Feasible relative improvement set). The feasible set in the relative improvement space is $\Omega(\mathcal{F}) = \{\boldsymbol{\rho} \in \mathbb{R}^m : \boldsymbol{\rho} = T(\boldsymbol{r})$ for some $\boldsymbol{r} \in \mathcal{R}(\mathcal{F})\} = T(\mathcal{R}(\mathcal{F}))$.

**Definition A.11** (Pareto frontier of relative improvement set). The Pareto frontier of the relative improvement set $\Omega(\mathcal{F})$ is $\mathcal{P}_\Omega(\mathcal{F}) = \{\boldsymbol{\rho} \in \Omega(\mathcal{F}) :$ there exists no $\boldsymbol{\rho}' \in \Omega(\mathcal{F})$ such that $\boldsymbol{\rho}' \succ_P \boldsymbol{\rho}\}$.

*Remark* A.12 (Properties of the transformation). The transformation $T$ is affine (specifically, a composition of translation and scaling on each coordinate). Therefore: (i) $T$ preserves convexity: if $\mathcal{R}(\mathcal{F})$ is convex, then $\Omega(\mathcal{F})$ is convex, (ii) $T$ preserves compactness: if $\mathcal{R}(\mathcal{F})$ is compact, then $\Omega(\mathcal{F})$ is compact, and (iii) Pareto optimality is preserved under $T$: if $\boldsymbol{r}$ is Pareto optimal in $\mathcal{R}(\mathcal{F})$, then $T(\boldsymbol{r})$ is Pareto optimal in $\Omega(\mathcal{F})$. Under Assumption 3.2 or Assumption 3.3, Theorem 3.4 establishes that $\mathcal{R}(\mathcal{F})$ is compact with all Pareto optimal points in $\mathcal{R}(\mathcal{F})$ rather than only in its convex hull. These properties transfer to $\Omega(\mathcal{F})$.

*Remark* A.13 (Connection to risk space Pareto frontier). The Pareto frontier in relative improvement space is precisely the image under $T$ of the Pareto frontier in risk space:

$$\mathcal{P}_\Omega(\mathcal{F}) = T(\mathcal{P}_\mathcal{R}(\mathcal{F}))$$

The direction of Pareto dominance reverses under the transformation: in risk space, lower values are preferred ($r'_g \leq r_g$), while in relative improvement space, higher values are preferred ($\rho'_g \geq \rho_g$).

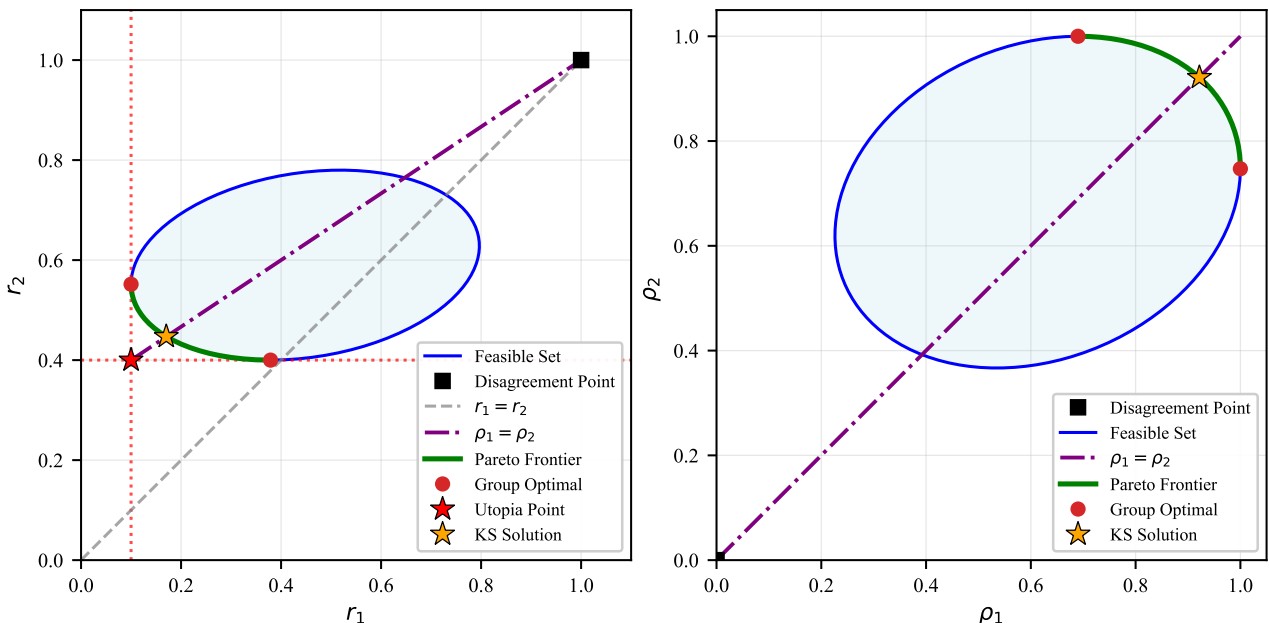

*Figure 6.* Transformation between risk space and relative improvement space. Left: Risk space with group-optimal risks $r_1^* = 0.1$, $r_2^* = 0.4$ (utopia point) and baseline disagreement point $(1, 1)$. Right: Relative improvement space where the disagreement point maps to $(0, 0)$ and group optimal points to $(1, \rho_2^{(1)})$ and $(\rho_1^{(2)}, 1)$. The fairness diagonal $\rho_1 = \rho_2$ intersects the Pareto frontier at the KS solution (orange star).

## B. Proof of Theoretical Results

### B.1. Bargaining Problem Perspective

We first establish formal equivalence and uniqueness results for the relative improvement optimization problem underlying Section 3.2. The equivalence result below (Theorem B.1, part 1.) is a direct adaptation of the Kalai–Smorodinsky solution to our setting. We include the proof for completeness and to make the connection explicit.

**Theorem B.1** (Equal Relative Improvement and Maximin Equivalence)**.** *For the two-group case ($m = 2$), under Assumption 3.2 or Assumption 3.3:*

1. *The constraint $\rho_1(f) = \rho_2(f)$ intersects the Pareto frontier at exactly one point.*

2. *This unique point coincides with the maximin solution $\max_{f \in \mathcal{F}} \min_{g \in \{1,2\}} \rho_g(f)$.*

**Lemma B.2** (Berge's Maximum Theorem (Berge, 1963))**.** *Let $X$ be a compact topological space and $\Theta$ a topological space. Let $C : \Theta \rightrightarrows X$ be a compact-valued correspondence with $C(\theta) \neq \emptyset$ for all $\theta \in \Theta$. Let $\xi : X \times \Theta \to \mathbf{R}$ be continuous. Define*

$$V(\theta) = \sup\{\xi(x, \theta) : x \in C(\theta)\}.$$

*If $C$ is continuous (upper and lower hemicontinuous), then $V$ is continuous.*

*Proof of Theorem B.1.* Under Assumption 3.2 or Assumption 3.3, parts (1) and (2) of Theorem 3.4 ensure that $\mathrm{conv}(\mathcal{R}(\mathcal{F}))$ is compact and convex, with all Pareto optimal points lying in $\mathcal{R}(\mathcal{F})$ rather than only in its convex hull. Since the transformation $T$ is affine, these properties transfer to $\Omega(\mathcal{F})$: the set $\mathrm{conv}(\Omega(\mathcal{F}))$ is compact and convex, and the Pareto frontiers of $\Omega(\mathcal{F})$ and $\mathrm{conv}(\Omega(\mathcal{F}))$ coincide. Therefore, without loss of generality, we may assume $\Omega(\mathcal{F})$ is compact and convex for the remainder of the proof.

Let $\boldsymbol{\rho}^{(1)} = (1, \rho_2^{(1)})$ and $\boldsymbol{\rho}^{(2)} = (\rho_1^{(2)}, 1)$ denote the group-optimal points. Specifically, define $\rho_2^{(1)} = \max\{\rho_2 : (1, \rho_2) \in \Omega(\mathcal{F})\}$ and $\rho_1^{(2)} = \max\{\rho_1 : (\rho_1, 1) \in \Omega(\mathcal{F})\}$. If $\rho_2^{(1)} = 1$ or $\rho_1^{(2)} = 1$, the result follows immediately. Otherwise, assume $\rho_2^{(1)} < 1$ and $\rho_1^{(2)} < 1$.

For a compact convex set $\Omega(\mathcal{F})$ in $\mathbf{R}^2$, the upper-right boundary (Pareto frontier) can be locally represented as the graph of a function. Specifically, for each $\rho_1 \in [\rho_1^{(2)}, 1]$, define

$$\phi(\rho_1) = \max\{\rho_2 : (\rho_1, \rho_2) \in \Omega(\mathcal{F})\}.$$

The maximum exists by compactness of $\Omega(\mathcal{F})$, and the resulting function $\phi : [\rho_1^{(2)}, 1] \to \mathbf{R}$ describes the upper boundary of feasible set (green curve in Figure 6). Also, the boundary of a convex compact set $\Omega(\mathcal{F})$ ensures that $C : [\rho_{1,\min}, \rho_{1,\max}] \rightrightarrows (-\infty, 1]$ such that $C(\rho_1) := \{\rho_2 : (\rho_1, \rho_2) \in \Omega(\mathcal{F})\} \neq \emptyset$ is a compact-valued correspondence. Denote $f(\rho_2, \rho_1) = \rho_2$ which is a continuous function; then by maximum theorem (Lemma B.2),

$$\phi(\rho_1) = \sup\{\rho_2 = f(\rho_2, \rho_1) : \rho_2 \in C(\rho_1)\}$$

is a continuous function.

For any $\rho_1 \in [\rho_1^{(2)}, 1]$, the point $(\rho_1, \phi(\rho_1))$ is Pareto optimal. If not, there would exist $(\rho_1', \rho_2') \in \Omega(\mathcal{F})$ with $\rho_1' \geq \rho_1$ and $\rho_2' \geq \phi(\rho_1)$, with at least one inequality strict. If $\rho_1' = \rho_1$, this contradicts the definition of $\phi(\rho_1)$ as the maximum. If $\rho_1' > \rho_1$, then

$$\rho_2 = \phi(\rho_1) \geq \frac{\phi(\rho_1^{(2)})(\rho_1' - \rho_1) + \phi(\rho_1')(\rho_1 - \rho_1^{(2)})}{\rho_1' - \rho_1^{(2)}}$$

$$= \frac{(\rho_1' - \rho_1) + \phi(\rho_1')(\rho_1 - \rho_1^{(2)})}{\rho_1' - \rho_1^{(2)}} > \phi(\rho_1') \geq \rho_2',$$

which is a contradiction.

Define $g(\rho_1) = \phi(\rho_1) - \rho_1$. Then,

$$g(\rho_1^{(2)}) = 1 - \rho_1^{(2)} > 0,$$
$$g(1) = \rho_2^{(1)} - 1 < 0.$$

By continuity of $\phi$ and the Intermediate Value Theorem, there exists $\rho_1^* \in (\rho_1^{(2)}, 1)$ such that $g(\rho_1^*) = 0$, i.e., $\phi(\rho_1^*) = \rho_1^*$ Geometrically, since one group's optimal point lies above and the other below the fairness diagonal, continuity of the frontier guarantees an intersection (see Figure 6).

Therefore, $(\rho_1^*, \rho_1^*)$ lies on both the Pareto frontier and the fairness diagonal, establishing part 1.

For part 2, we show that this point is the maximin solution. Because of Pareto Optimality of $(\rho_1^*, \rho_1^*)$, any other point $(\rho_1, \rho_2) \in \Omega(\mathcal{F})$ must have $\rho_1 \leq \rho_1^*$ or $\rho_2 \leq \rho_1^*$ (or both). Therefore, $\min\{\rho_1, \rho_2\} \leq \rho_1^* = \min\{\rho_1^*, \rho_1^*\}$, which shows that $(\rho_1^*, \rho_1^*)$ maximizes the minimum relative improvement. $\qquad\square$

This result strengthens the connection between our fairness framework and the KS solution: the KS-solution, which selects the point on the Pareto frontier where both players achieve equal normalized gains, coincides with the maximin relative improvement solution.

We next establish a uniqueness result for the predictor, clarifying the additional assumptions required for the relative improvement solution to be uniquely realized. As discussed in Remark 3.1, this result is strictly stronger than uniqueness of the induced risk vector or of the bargaining solution itself.

**Theorem B.3** (Uniqueness of Maximin Solution). *Under Assumption 3.2 or Assumption 3.3 with strict convexity of the loss function, the maximin fairness optimization problem*

$$\max_{f \in \mathcal{F}} \min_{g \in \{1, \ldots, m\}} \rho_g(f)$$

*admits a unique solution.*

*Proof of Theorem B.3.* Define the worst-group relative improvement function

$$F(f) := \min_{g \in \{1,\dots,m\}} \rho_g(f) = \min_{g \in \{1,\dots,m\}} \frac{r_g^0 - R_g(f)}{r_g^0 - r_g^*}.$$

We show that $F$ is strictly concave. For any $f_1, f_2 \in \mathcal{F}$ with $f_1 \neq f_2$ and $\lambda \in (0,1)$, let $f_\lambda = \lambda f_1 + (1-\lambda)f_2$. By strict convexity of each risk function $R_g$, $\rho_g$ is strictly concave and

$$\rho_g(f_\lambda) > \lambda \rho_g(f_1) + (1-\lambda)\rho_g(f_2)$$

for each $g \in \{1, \dots, m\}$. Taking the minimum over groups,

$$F(f_\lambda) = \min_{g \in \{1,\dots,m\}} \rho_g(f_\lambda) > \min_{g \in \{1,\dots,m\}} \{\lambda \rho_g(f_1) + (1-\lambda)\rho_g(f_2)\}$$
$$\geq \lambda \min_{g \in \{1,\dots,m\}} \rho_g(f_1) + (1-\lambda) \min_{g \in \{1,\dots,m\}} \rho_g(f_2) = \lambda F(f_1) + (1-\lambda)F(f_2).$$

Thus $F$ is strictly concave on the convex set $\mathcal{F}$. Under our assumptions, $F$ is continuous and $\mathcal{F}$ (or its parameterization $\Theta$) is compact, so a maximizer exists. Strict concavity ensures uniqueness: if there were two distinct maximizers $f_1 \neq f_2$ with $F(f_1) = F(f_2) = F^*$, then $F(f_{1/2}) > F^*$, contradicting maximality. $\square$

Moreover, when the loss is strictly convex, the maximin and leximin solutions coincide. This follows from the fact that every leximin solution is, by definition, a maximin solution, while the converse does not hold in general. If the maximin solution is unique, it must therefore also be the leximin solution. Under strict convexity of the loss, the maximin relative improvement solution satisfies the axioms discussed in Proposition 4.4. In the absence of strict convexity, there may exist multiple predictors achieving the same maximin relative improvement value; in this case, the leximin solutions satisfy the axioms in Proposition 4.4.

We provide the proof of the following result regarding the properties of the feasible risk set in Section 3.4.

**Proposition B.4.** *For the parametric setting, Assumption 3.2(ii) can be verified if the loss satisfies a Lipschitz condition with integrable Lipschitz constant. For the RKHS setting with $\mathcal{F} = \{f \in \mathcal{H} : \|f\|_{\mathcal{H}} \leq R\}$, Assumption 3.3(ii) holds under a standard $p$-growth condition $|\ell(y,t)| \leq C(1+|t|^p)$ combined with a moment condition on the kernel $\mathbf{E}_g[k(X,X)^{p/2}] < \infty$.*

*Proof of Proposition B.4.*

**Parametric Setting - Lipschitz Condition.** Suppose the loss function satisfies a Lipschitz condition: there exists $L : \mathcal{X} \times \mathcal{Y} \to \mathbf{R}_+$ such that
$$|\ell(y, f_{\theta_1}(x)) - \ell(y, f_{\theta_2}(x))| \leq L(x,y)\|\theta_1 - \theta_2\|$$
for all $\theta_1, \theta_2 \in \Theta$ with $\mathbf{E}_g[L(X,Y)] < \infty$ for all $g \in \mathcal{G}$. Then for any $\theta_0 \in \Theta$, we can construct the dominating function

$$g(x,y) = |\ell(y, f_{\theta_0}(x))| + \text{diam}(\Theta) \cdot L(x,y),$$

where $\text{diam}(\Theta) = \sup_{\theta_1, \theta_2 \in \Theta} \|\theta_1 - \theta_2\| < \infty$ by compactness of $\Theta$. For any $\theta \in \Theta$,

$$|\ell(y, f_\theta(x))| \leq |\ell(y, f_{\theta_0}(x))| + |\ell(y, f_\theta(x)) - \ell(y, f_{\theta_0}(x))|$$
$$\leq |\ell(y, f_{\theta_0}(x))| + L(x,y)\|\theta - \theta_0\|$$
$$\leq |\ell(y, f_{\theta_0}(x))| + L(x,y) \cdot \text{diam}(\Theta)$$
$$= h(x,y).$$

Since $\mathbf{E}_g[L(X,Y)] < \infty$ and $|\ell(y, f_{\theta_0}(x))|$ is integrable, we have $\mathbf{E}_g[h(X,Y)] < \infty$ for all $g \in \mathcal{G}$, verifying Assumption 3.2(2).

**RKHS Setting - $p$-Growth Condition.** Suppose the loss function satisfies a $p$-growth condition: there exist constants $C > 0$ and $p \geq 1$ such that
$$|\ell(y,t)| \leq C(1+|t|^p) \quad \text{for all } (y,t) \in \mathcal{Y} \times \mathbf{R}.$$

Since $\mathcal{H}$ is a reproducing kernel Hilbert space, every $f \in \mathcal{F} \subset \mathcal{H}$ satisfies the reproducing property

$$|f(x)| \le \|f\|_{\mathcal{H}} \sqrt{k(x,x)}.$$

Therefore, for all $f \in \mathcal{F}$,

$$
\begin{aligned}
|\ell(y, f(x))| &\le C(1 + |f(x)|^p) \\
&\le C(1 + \|f\|_{\mathcal{H}}^p \cdot k(x,x)^{p/2}) \\
&\le C(1 + B^p \cdot k(x,x)^{p/2}) =: h(x,y),
\end{aligned}
$$

where $B = \sup_{f \in \mathcal{F}} \|f\|_{\mathcal{H}} < \infty$. The finiteness of $B$ follows from weak compactness of $\mathcal{F}$ in $\mathcal{H}$: a weakly compact set in a Hilbert space is norm-bounded. If the kernel moment condition holds, i.e., $\mathbf{E}_g[k(X,X)^{p/2}] < \infty$ for all $g \in \mathcal{G}$, then

$$\mathbf{E}_g[h(X,Y)] = C(1 + B^p \cdot \mathbf{E}_g[k(X,X)^{p/2}]) < \infty,$$

verifying Assumption 3.3(2). $\qquad\square$

This proposition shows that Assumptions 3.2(2) and 3.3(2) are satisfied under standard regularity conditions in commonly used parametric and RKHS settings.

We now prove the theorem linking the fair learning formulation with the corresponding bargaining problem.

*Proof of Theorem 3.4.* We prove the result for both the parametric setting (Assumption 3.2) and the nonparametric setting (Assumption 3.3).

**Continuity and Convexity of Risk Functions.**

*Parametric case.* For any $\theta_n \to \theta$ in $\Theta$, Assumption 3.2(1) gives $\ell(y, f_{\theta_n}(x)) \to \ell(y, f_\theta(x))$ pointwise. By Assumption 3.2(2), $|\ell(y, f_{\theta_n}(x))| \le h(x,y)$ where $\mathbf{E}_g[h(X,Y)] < \infty$. The Dominated Convergence Theorem implies $R_g(\theta_n) \to R_g(\theta)$, so $R_g$ is continuous.

For convexity, if $0 < \lambda < 1$, then by convexity of $\ell$ in $\theta$,

$$R_g(\lambda\theta_1 + (1-\lambda)\theta_2) = \mathbf{E}_g[\ell(Y, f_{\lambda\theta_1 + (1-\lambda)\theta_2}(X))] \le \lambda R_g(\theta_1) + (1-\lambda)R_g(\theta_2).$$

*Nonparametric case.* Let $(f_n) \subset \mathcal{F}$ with $f_n \to f$ in the topology $\tau$. By the continuity of evaluation maps, $f_n(x) \to f(x)$ for every $x$. Thus $\ell(Y, f_n(X)) \to \ell(Y, f(X))$ almost surely by continuity of $\ell$. With the dominating function from Assumption 3.3(2), the Dominated Convergence Theorem gives $R_g(f_n) \to R_g(f)$.

For convexity in the nonparametric case, if $\ell(y,t)$ is convex in $t$, then for $0 < \lambda < 1$,

$$R_g(\lambda f_1 + (1-\lambda)f_2) \le \lambda R_g(f_1) + (1-\lambda)R_g(f_2).$$

**(1) Compactness of $\mathcal{R}(\mathcal{F})$.**

*Parametric case.* The map $\boldsymbol{R} : \Theta \to \mathbb{R}^m$ defined by $\boldsymbol{R}(\theta) = (R_1(\theta), \dots, R_m(\theta))$ is continuous. Since $\Theta$ is compact by Assumption 3.2(3), the image $\mathcal{R}(\mathcal{F}) = R(\Theta)$ is compact.

*Nonparametric case.* The map $\boldsymbol{R} : \mathcal{F} \to \mathbb{R}^m$ is continuous with respect to the topology $\tau$ on $\mathcal{F}$. Since $\mathcal{F}$ is weakly compact by Assumption 3.3(3), the image $\mathcal{R}(\mathcal{F}) = R(\mathcal{F})$ is compact.

**(2) No Pareto optimal points in conv$(\mathcal{R}(\mathcal{F})) \setminus \mathcal{R}(\mathcal{F})$.**

Let $\boldsymbol{r} \in \text{conv}(\mathcal{R}(\mathcal{F})) \setminus \mathcal{R}(\mathcal{F})$. Then there exist $k \ge 2$ parameters/functions and weights $\lambda_i > 0$ with $\sum_i \lambda_i = 1$ such that $\boldsymbol{r} = \sum_{i=1}^k \lambda_i \boldsymbol{R}(\theta_i)$ (or $\boldsymbol{R}(f_i)$).

Define $\theta^* = \sum_{i=1}^{k} \lambda_i \theta_i$ (or $f^* = \sum_{i=1}^{k} \lambda_i f_i$), which lies in $\Theta$ (or $\mathcal{F}$) by convexity. By convexity of each $R_g$,

$$R_g(\theta^*) \leq \sum_{i=1}^{k} \lambda_i R_g(\theta_i) = r_g$$

for all $g$. Since $\boldsymbol{r} \notin \mathcal{R}(\mathcal{F})$ but $\boldsymbol{R}(\theta^*) \in \mathcal{R}(\mathcal{F})$, we have $\boldsymbol{R}(\theta^*) \neq \boldsymbol{r}$. Combined with $\boldsymbol{R}(\theta^*) \leq \boldsymbol{r}$ componentwise, this means at least one inequality is strict, so $\boldsymbol{R}(\theta^*)$ strictly dominates $\boldsymbol{r}$ in at least one component. Therefore, $\boldsymbol{r}$ cannot be Pareto optimal.

**Additional properties under strict convexity.**

When the loss function is strictly convex in $\theta$ (or $t$), the convexity inequalities in Step 1 become strict inequalities whenever $\theta_1 \neq \theta_2$ (or $f_1(X) \neq f_2(X)$ with positive probability). This strict convexity yields two additional properties:

**(3) Weakly Pareto optimal implies Pareto optimal.**

Suppose there exist parameters/functions $\theta_1, \theta_2$ (or $f_1, f_2$) such that $R_g(\theta_1) \leq R_g(\theta_2)$ for all $g$ with strict inequality for at least one $g$.

Define $\theta' = \frac{\theta_1 + \theta_2}{2}$ (or $f' = \frac{f_1 + f_2}{2}$), which lies in $\Theta$ (or $\mathcal{F}$) by convexity. By strict convexity,

$$R_g(\theta') < \frac{1}{2}R_g(\theta_1) + \frac{1}{2}R_g(\theta_2) \leq R_g(\theta_2)$$

for all $g$. Thus $\theta_2$ (or $f_2$) is not weakly Pareto optimal, proving that every weakly Pareto optimal point must be Pareto optimal.

**(4) No weakly Pareto optimal points in $\mathrm{conv}(\mathcal{R}(\mathcal{F})) \setminus \mathcal{R}(\mathcal{F})$.**

Let $\boldsymbol{r} \in \mathrm{conv}(\mathcal{R}(\mathcal{F})) \setminus \mathcal{R}(\mathcal{F})$. As in part (2), there exist $k \geq 2$ parameters/functions and weights $\lambda_i > 0$ with $\sum_i \lambda_i = 1$ such that $\boldsymbol{r} = \sum_{i=1}^{k} \lambda_i \boldsymbol{R}(\theta_i)$, and $\theta^* = \sum_{i=1}^{k} \lambda_i \theta_i$ lies in $\Theta$ (or $\mathcal{F}$).

By strict convexity,

$$R_g(\theta^*) < \sum_{i=1}^{k} \lambda_i R_g(\theta_i) = r_g$$

for all $g$, so $\boldsymbol{R}(\theta^*)$ strictly dominates $\boldsymbol{r}$ in all components. Therefore, $\boldsymbol{r}$ is not weakly Pareto optimal. $\qquad\square$

**B.2. Fairness Guarantees of Relative Improvement**

We provide proofs of the fairness guarantees for relative improvement stated in the main text Section 4.

*Proof of Theorem 4.1.* Since $f_{\mathrm{RI}}$ maximizes the minimum relative improvement,

$$\min_{g \in \mathcal{G}} \rho_g(f_{\mathrm{RI}}) = \max_{f \in \mathcal{F}} \min_{g \in \mathcal{G}} \rho_g(f) = \max_{f \in \mathcal{F}} \min_{g \in \mathcal{G}} \frac{R_g(f_0) - R_g(f)}{R_g(f_0) - R_g(f_g^*)}$$
$$\geq \min_{g \in \mathcal{G}} \frac{R_g(f_0) - R_g(f_0)}{R_g(f_0) - R_g(f_g^*)} = 0,$$

where the inequality follows from feasibility of $f_0$. Hence $\rho_g(f_{\mathrm{RI}}) \geq 0$ for every group $g$, which by definition of relative improvement implies $R_g(f_{\mathrm{RI}}) \leq R_g(f_0)$. $\qquad\square$

*Proof of Proposition 4.2.* Each axiom follows by direct verification, adapting Kalai & Smorodinsky (1975) to our framework where groups correspond to players and negative risks to utilities. $\qquad\square$

*Proof of Theorem 4.3.* Let $S = \mathcal{R}(\mathcal{F})$ and let $\mathrm{comp}(S)$ denote its comprehensive closure. We show that the leximin solution over $S$ coincides with that over $\mathrm{comp}(S)$.

First, observe that any point $r' \in \text{comp}(S) \setminus S$ is Pareto dominated by some point $r \in S$. Indeed, by definition of the comprehensive closure, there exists $r \in S$ such that $r_g \leq r'_g$ for all $g$, with strict inequality for at least one group. Hence, $r'$ cannot be Pareto optimal in $\text{comp}(S)$.

Second, the leximin solution is Pareto optimal. Therefore, no point in $\text{comp}(S) \setminus S$ can be selected by the leximin criterion, and any leximin solution in $\text{comp}(S)$ must lie in $S$.

Finally, since $f_0 \in \mathcal{F}$, the baseline risk vector $-d$ belongs to $S$. By Theorem 4.1, a leximin relative improvement solution satisfies $R_g(f_{\text{RI}}) \leq R_g(f_0)$ for all $g$, implying that the leximin solution lies within the bounds defining $\text{comp}(S)$.

Combining these observations, we conclude that the leximin solution over $S$ coincides with that over $\text{comp}(S)$. $\qquad\square$

*Proof of Proposition 4.4.* The leximin solution on a feasible set $S$ equals the leximin solution on its comprehensive closure $\text{comp}(S)$. Denote by $f(S, d)$ the leximin solution when $S$ is a feasible set with disagreement point $d$, and by $g(S, d)$ the leximin solution when $S$ is a comprehensive feasible set with disagreement point $d$. Then $f(S, d) = g(\text{comp}(S), d)$.

Since $g(\cdot, \cdot)$ satisfies the five axioms on comprehensive sets (Imai, 1983):

1. **Pareto Optimality (PO):**
$$f(S, d) = g(\text{comp}(S), d) \in \text{PF}(\text{comp}(S)) = \text{PF}(S).$$

2. **Symmetry (SYM):** For any permutation $\pi$,
$$\pi * f(S, d) = \pi * g(\text{comp}(S), d) = g(\pi * \text{comp}(S), \pi * d)$$
$$= g(\text{comp}(\pi * S), \pi * d) = f(\pi * S, \pi * d).$$

3. **Scale Invariance (SI)** For any affine transformation $T$ applied coordinatewise,
$$T(f(S, d)) = T(g(\text{comp}(S), d)) = g(T(\text{comp}(S)), T(d))$$
$$= g(\text{comp}(T(S)), T(d)) = f(T(S), T(d)).$$

4. **Independence of Irrelevant Alternatives with Ideal point (IIIA)** If $\text{comp}(S_1) \subseteq \text{comp}(S_2)$ with the same ideal point $I(\text{comp}(S_1)) = I(\text{comp}(S_2))$, and if $f(S_2, d) = g(\text{comp}(S_2), d) \in \text{comp}(S_1)$, then
$$f(S_1, d) = g(\text{comp}(S_1), d) = g(\text{comp}(S_2), d) = f(S_2, d).$$

5. **Modified Individual Monotonicity (IM')** If $\text{comp}(S_1) \subseteq \text{comp}(S_2)$ and the relevant projections for player $i$ coincide, $^i\text{comp}(S_1) = {}^i\text{comp}(S_2)$ (equivalently, $\text{comp}(^iS_1) = \text{comp}(^iS_2)$), then
$$f_i(S_1, d) = g_i(\text{comp}(S_1), d) \leq g_i(\text{comp}(S_2), d) = f_i(S_2, d).$$

Our relative improvement maximizer $f_{\text{RI}}$ operates on function classes $\mathcal{F}$ rather than abstract feasible sets $S$ under the mapping in Section 3.1. $\qquad\square$

### B.3. Empirical Estimator

We establish concentration inequalities and convergence rates for the empirical estimator in Section 5.

*Proof of Lemma 5.2.* (A) By symmetrization (Lemma 2.3.1 in (Van der Vaart, 2000)),
$$\mathbb{E}\left[\sup_{f \in \mathcal{F}} \left|\widehat{R}_g(f) - R_g(f)\right|\right] \leq \frac{2}{n_g} \mathbb{E}_{X, Y, \varepsilon}\left[\sup_{f \in \mathcal{F}} \left|\sum_{i=1}^{n_g} \varepsilon_i \, \ell(Y_i, f(X_i))\right|\right]$$
$$= 2\mathcal{R}_{n_g}(\ell \circ \mathcal{F}),$$

where $\mathcal{R}_{n_g}(\ell \circ \mathcal{F})$ is the Rademacher complexity of the function class $\ell \circ \mathcal{F}$ when the sample size is $n_g$. By Dudley's entropy integral bound (Theorem 5.22 in (Wainwright, 2019)),

$$\mathcal{R}_{n_g}(\ell \circ \mathcal{F}) = \frac{1}{\sqrt{n_g}} \mathbb{E}_\varepsilon \left[ \sup_{f \in \mathcal{F}} Z_f \right] \leq \frac{32}{\sqrt{n_g}} \int_0^1 \sqrt{\log \mathcal{N}(u; \ell \circ \mathcal{F}, L_2(P_g))} \, du. \tag{20}$$

Since $\ell(\cdot, \cdot)$ is L-Lipschitz in the second argument, for all $f, f' \in \mathcal{F}$,

$$\|\ell(\cdot, f(\cdot)) - \ell(\cdot, f'(\cdot))\|_{L_2(P_g)} \leq L \|f - f'\|_{L_2(P_{g,X})}.$$

Therefore, $\mathcal{N}(u; \ell \circ \mathcal{F}, L_2(P_g)) \leq \mathcal{N}(u/L; \mathcal{F}, L_2(P_{g,X}))$, and we obtain

$$\mathcal{R}_{n_g}(\ell \circ \mathcal{F}) \leq \frac{32}{\sqrt{n_g}} \int_0^1 \sqrt{\log \mathcal{N}(u/L; \mathcal{F}, L_2(P_{g,X}))} \, du.$$

Changing the variables by $t = u/L$,

$$\int_0^1 \sqrt{\log \mathcal{N}(u/L; \mathcal{F}, L_2(P_{g,X}))} \, du = L \int_0^{1/L} \sqrt{\log \mathcal{N}(t; \mathcal{F}, L_2(P_{g,X}))} \, dt$$

Under the entropy condition,

$$\int_0^{1/L} \sqrt{\log \mathcal{N}(t; \mathcal{F}, L_2(P_{g,X}))} \, dt \leq \sqrt{C_0} \int_0^{1/L} t^{-p/2} \, dt$$

$$= \frac{\sqrt{C_0}}{1 - p/2} \left( \frac{1}{L} \right)^{1 - \frac{p}{2}} < \infty.$$

Therefore,

$$\mathbb{E} \left[ \sup_{f \in \mathcal{F}} \left| \hat{R}_g(f) - R_g(f) \right| \right] \leq 2 \mathbb{E} \left[ \widehat{\mathcal{R}}_{n_g}(\ell \circ \mathcal{F}) \right] \leq C_1 \frac{L}{\sqrt{n_g}}. \tag{21}$$

Define

$$\Phi(S) := \sup_{f \in \mathcal{F}} \left| \hat{R}_g(f) - R_g(f) \right| = \sup_{f \in \mathcal{F}} \left| \frac{1}{n_g} \sum_{i \in I_g} \left( \ell(y_i, f(x_i)) - \mathbb{E}[\ell(Y, f(X))] \right) \right|,$$

as a function of the sample $S = \{(x_i, y_i)\}_{i \in I_g}$. If we replace a single point $(x_i, y_i)$ by $(x_i', y_i')$, then by boundedness $\ell \in [0, 1]$,

$$|\Phi(S) - \Phi(S^{(i)})| \leq \frac{1}{n_g} \sup_{f \in \mathcal{F}} |\ell(y_i, f(x_i)) - \ell(y_i', f(x_i'))| \leq \frac{1}{n_g}.$$

Thus $\Phi$ satisfies the bounded-differences condition with $c_i = 1/n_g$. By McDiarmid's inequality (Corollary 2.21 in (Wainwright, 2019)), for any $t > 0$,

$$\mathbb{P} \left\{ \Phi(S) - \mathbb{E}[\Phi(S)] \geq t' \right\} \leq \exp \left( -\frac{2t'^2}{\sum_i c_i^2} \right) = \exp(-2n_g t'^2).$$

Choosing $t' = \frac{t}{\sqrt{n_g}}$ gives, with probability at least $1 - 2e^{-t}$,

$$\sup_{f \in \mathcal{F}} \left| \hat{R}_g(f) - R_g(f) \right| \leq \mathbb{E} \left[ \sup_{f \in \mathcal{F}} \left| \hat{R}_g(f) - R_g(f) \right| \right] + \sqrt{\frac{t}{n_g}}. \tag{22}$$

Combining (21) and (22) yields the desired bound.

(B) Since we no longer have a bound on the loss function, consider the bounded loss function defined as

$$\ell_f = \phi_\tau(\ell_f) + (\ell_f - \phi_\tau(\ell_f)), \qquad \phi_\tau(u) = \text{sign}(u) \min\{|u|, \tau\},$$

where $\ell_f(x, y) := \ell(y, f(x))$ and $\tau > 0$ is some constant. The remainder term satisfies

$$|\ell_f - \phi_\tau(\ell_f)| \leq F_g(x, y)\,\mathbf{1}\{F_g(x, y) \geq \tau\} =: Z(x, y).$$

Hence, the tail contribution to the empirical deviation obeys

$$\sup_{f \in \mathcal{F}} \left|(\hat{P}_g - P_g)(\ell_f - \phi_\tau(\ell_f))\right| \leq \left|(\hat{P}_g - P_g)Z\right|.$$

Since $F_g(X, Y)$ is sub-Gaussian with parameter $\sigma$,

$$\mathbb{E}[Z^2] = \mathbb{E}\big[F_g^2\,\mathbf{1}\{F_g \geq \tau\}\big] = \int_\tau^\infty 2t\, P(F_g > t)\, dt \leq \int_\tau^\infty 2t\, e^{-t^2/(2\sigma^2)}\, dt = 2\sigma^2 e^{-\tau^2/(2\sigma^2)}.$$

Thus,

$$\mathbb{E}\Big[\sup_{f \in \mathcal{F}} \big|(\hat{P}_g - P_g)(\ell_f - \phi_\tau(\ell_f))\big|\Big] \leq \frac{C_3\sigma}{\sqrt{n_g}}\, e^{-\tau^2/(2\sigma^2)}.$$

Moreover, since $(\hat{P}_g - P_g)Z$ is $\sigma/\sqrt{n_g}$-sub-Gaussian, it follows that $\sup_{f \in \mathcal{F}} \big|(\hat{P}_g - P_g)(\ell_f - \phi_\tau(\ell_f))\big|$ is also $\sigma/\sqrt{n_g}$-sub-Gaussian. Therefore, by concentration of sub-Gaussian random variables, with probability at least $1 - 2e^{-t}$,

$$\sup_{f \in \mathcal{F}} \big|(\hat{P}_g - P_g)(\ell_f - \phi_\tau(\ell_f))\big| \leq \mathbb{E}\Big[\sup_{f \in \mathcal{F}} \big|(\hat{P}_g - P_g)(\ell_f - \phi_\tau(\ell_f))\big|\Big] + C_4\sigma\sqrt{\frac{t}{n_g}}$$

$$\leq \frac{C_3\sigma}{\sqrt{n_g}}\, e^{-\tau^2/(2\sigma^2)} + C_4\sigma\sqrt{\frac{t}{n_g}}. \tag{23}$$

On the other hand, since $\phi_\tau(\ell_f)$ is a bounded and Lipschitz loss satisfying the entropy condition, it follows from part (A) that, with probability at least $1 - 2e^{-t}$,

$$\sup_{f \in \mathcal{F}} \big|(\hat{P}_g - P_g)\phi_\tau(\ell_f)\big| \leq \frac{C_1 L}{\sqrt{n_g}} + C_2\sqrt{\frac{t}{n_g}}. \tag{24}$$

Combining (23) and (24) via the union bound, we obtain that with probability at least $1 - 4e^{-t}$,

$$\sup_{f \in \mathcal{F}} \big|(\hat{P}_g - P_g)\ell_f\big| \leq \sup_{f \in \mathcal{F}} \big|(\hat{P}_g - P_g)\phi_\tau(\ell_f)\big| + \sup_{f \in \mathcal{F}} \big|(\hat{P}_g - P_g)(\ell_f - \phi_\tau(\ell_f))\big|$$

$$\leq \frac{C_1 L}{\sqrt{n_g}} + C_2\sqrt{\frac{t}{n_g}} + \frac{C_3\sigma}{\sqrt{n_g}}\, e^{-\tau^2/(2\sigma^2)} + C_4\sigma\sqrt{\frac{t}{n_g}}$$

$$\leq \frac{C_1' L}{\sqrt{n_g}} + C_2'\sigma\sqrt{\frac{t}{n_g}},$$

where $C_1' = C_1 + C_3 e^{-\tau^2/(2\sigma^2)}$ and $C_2' = C_2 + C_4$. $\qquad\square$

*Proof of Theorem 5.4.* Fix $\delta \in (0, 1)$ and set

$$\varepsilon_0 := \max_{g \in \mathcal{G}} r_{n_g}\big(\log(2m/\delta)\big).$$

By Assumption 5.1 and a union bound over $g \in \mathcal{G}$, with probability at least $1 - \delta$ we have, simultaneously for all $g$,

$$\sup_{f \in \mathcal{F}} \big|\hat{R}_g(f) - R_g(f)\big| \leq r_{n_g}\big(\log(2m/\delta)\big) \leq \varepsilon_0. \tag{25}$$

Consider the difference between the empirical and population relative improvements for a generic $f$:

$$\hat{\rho}_g(f) - \rho_g(f) = \frac{\hat{R}_g(f_0) - \hat{R}_g(f)}{\hat{R}_g(f_0) - \hat{R}_g^*} - \frac{R_g(f_0) - R_g(f)}{R_g(f_0) - R_g^*}.$$

For simplicity, write $A_g(f) = R_g(f_0) - R_g(f)$ and $A_g^* = A_g(f_g^*)$. Define the empirical counterparts $\hat{A}_g(f) = \hat{R}_g(f_0) - \hat{R}_g(f)$ and $\hat{A}_g^* = \hat{A}_g(f_g^*)$. Then

$$
|\hat{\rho}_g(f) - \rho_g(f)| = \left| \frac{\hat{A}_g(f)}{\hat{A}_g^*} - \frac{A_g(f)}{A_g^*} \right|
$$

$$
= \frac{|A_g^*(\hat{A}_g(f) - A_g(f)) - A_g(f)(\hat{A}_g^* - A_g^*)|}{\hat{A}_g^* A_g^*}
$$

$$
\leq \frac{|\hat{A}_g(f) - A_g(f)|}{\hat{A}_g^*} + \frac{A_g(f)}{A_g^*} \frac{|\hat{A}_g^* - A_g^*|}{\hat{A}_g^*}.
$$

Since $A_g^* = R_g(f_0) - R_g(f_g^*) \geq \Delta$ (Assumption 5.3), and by (25),

$$
|\hat{A}_g^* - A_g^*| \leq 2 \sup_{f \in \mathcal{F}} |\hat{R}_g(f) - R_g(f)| \leq 2\varepsilon_0 < \frac{\Delta}{2}.
$$

Hence, with probability at least $1 - \delta$, we have $\hat{A}_g^* \geq \Delta/2$ for all $g \in \mathcal{G}$. Moreover,

$$
|\hat{\rho}_g(f) - \rho_g(f)| \leq \frac{1}{\Delta} |\hat{A}_g(f) - A_g(f)| + \frac{2}{\Delta} |\hat{A}_g^* - A_g^*|
$$

$$
\leq \frac{3}{\Delta} \sup_{f \in \mathcal{F}} |\hat{R}_g(f) - R_g(f)| + \frac{3}{\Delta} |\hat{R}_g(f_0) - R_g(f_0)|.
$$

Therefore,

$$
\sup_{f \in \mathcal{F}} |\hat{\rho}_g(f) - \rho_g(f)| \leq \frac{3}{\Delta} \sup_{f \in \mathcal{F}} |\hat{R}_g(f) - R_g(f)| + \frac{3}{\Delta} |\hat{R}_g(f_0) - R_g(f_0)| \leq \frac{6}{\Delta} \varepsilon_0 =: \varepsilon.
$$

Also, plugging $f = \hat{f}_g^*$ into the display above gives

$$
\frac{\hat{A}_g(\hat{f}_g^*)}{\hat{A}_g(f_g^*)} \leq \frac{A_g(\hat{f}_g^*)}{A_g(f_g^*)} + \varepsilon \leq 1 + \varepsilon, \tag{26}
$$

$$
\frac{1}{\hat{A}_g(\hat{f}_g^*)} \geq (1 - \varepsilon) \frac{1}{\hat{A}_g(f_g^*)}. \tag{27}
$$

Now, with probability at least $1 - \delta$, for any $f \in \mathcal{F}$,

$$
\min_{g \in \mathcal{G}} \rho_g(\hat{f}_{\mathrm{RI}}) = \min_{g \in \mathcal{G}} \left[ \frac{A_g(\hat{f}_{\mathrm{RI}})}{A_g(f_g^*)} \right]
$$

$$
\geq \min_{g \in \mathcal{G}} \left[ \frac{\hat{A}_g(\hat{f}_{\mathrm{RI}})}{\hat{A}_g(f_g^*)} - \varepsilon \right]
$$

$$
\geq \min_{g \in \mathcal{G}} \left[ \frac{\hat{A}_g(\hat{f}_{\mathrm{RI}})}{\hat{A}_g(\hat{f}_g^*)} - \varepsilon \right]
$$

$$
\geq \min_{g \in \mathcal{G}} \left[ \frac{\hat{A}_g(f)}{\hat{A}_g(\hat{f}_g^*)} - \varepsilon \right]
$$

$$
\geq \min_{g \in \mathcal{G}} \left[ (1 - \varepsilon) \frac{\hat{A}_g(f)}{\hat{A}_g(f_g^*)} - \varepsilon \right]
$$

$$
\geq \min_{g \in \mathcal{G}} \left[ (1 - \varepsilon) \left\{ \frac{A_g(f)}{A_g(f_g^*)} - \varepsilon \right\} - \varepsilon \right]
$$

$$
\geq \min_{g \in \mathcal{G}} \left[ \frac{A_g(f)}{A_g(f_g^*)} - 3\varepsilon \right] = \min_{g \in \mathcal{G}} \rho_g(f) - 3\varepsilon.
$$

Therefore,

$$\min_{g \in \mathcal{G}} \rho_g(\hat{f}_{\text{RI}}) \;\geq\; \min_{g \in \mathcal{G}} \rho_g(f_{\text{RI}}) \;-\; \frac{C}{\Delta}\left(\max_{g \in \mathcal{G}} r_{n_g}\big(\log(2m/\delta)\big)\right),$$

which holds for some constant $C > 0$. $\hfill\square$

## C. Choice and Role of the Baseline Predictor

The baseline predictor $f_0$ plays two roles in our framework: it defines the disagreement point $d_g = -R_g(f_0)$ in the bargaining problem, and it serves as the reference against which relative improvement is measured.

**Interpretable default choices.** In most applications, $f_0$ should be chosen as an interpretable baseline that represents the trivial prediction made without access to covariates. In regression, the natural choice is the unconditional mean $f_0(x) = \mathbb{E}[Y]$, equivalently $f_0(x) = 0$ after centering. Under squared loss, the denominator becomes

$$R_g(f_0) - R_g(f_g^*) = \mathbb{E}_{P_g}[Y^2] - \sigma_g^2 = \|\beta_g\|_{\Sigma_g}^2,$$

which is the group-specific explained variance, and the relative improvement $\rho_g(f)$ coincides with the ratio of explained variance. Furthermore, it admits an equivalent interpretation as the ratio of coefficients of determination $R^2$:

$$\rho_g(\beta) = \frac{R_g^2(\beta)}{R_g^2(\beta_g)},$$

where $R_g^2(\beta)$ denotes the coefficient of determination for group $g$. This interpretability requirement is not merely a convention: it ensures that $R_g(f_0) - R_g(f_g^*)$ has a clear meaning as the total available signal for group $g$, and that $\rho_g(f)$ can be understood as the fraction of that signal captured by $f$. The choice $f_0(x) = 0$ is also consistent with the baseline used by Meinshausen & Bühlmann (2015). In binary classification, the majority-class predictor $f_0(x) = \pi_0$, where $\pi_0 = P(Y = 1)$, serves the same role. In some applications, $f_0$ may represent an existing deployed model with access to fewer covariates than $\mathcal{F}$, provided it admits a clear interpretation as a pre-intervention baseline; in this case, $R_g(f_0) - R_g(f_g^*)$ quantifies the additional predictability gained by incorporating the full covariate set.

**Bargaining interpretation.** In the bargaining formulation, $f_0$ represents the disagreement point: the outcome groups revert to if negotiation fails. Crucially, $f_0$ is exogenous to the bargaining problem—it is fixed before negotiation begins and is not itself a product of optimization over $\mathcal{F}$. This further motivates the interpretability requirement. A predictor that already embodies a compromise across groups—such as a pooled ERM solution or any Pareto-optimal predictor—is generally unsuitable as $f_0$ for two reasons. First, $R_g(f_0) - R_g(f_g^*)$ loses its interpretability as a measure of available signal, since it conflates the available signal with the effects of a prior modeling choice. Second, a group that is already satisfied with the existing compromise would have no incentive to participate in the bargaining process, undermining the cooperative framing.

**Sensitivity to the baseline choice.** Since $\rho_g(f)$ is defined relative to $f_0$, the solution $f_{\text{RI}}$ generally depends on the baseline. This is inherent to the formulation: group-wise improvement is meaningful only relative to a specified starting point, and changing $f_0$ defines a different bargaining problem rather than a robustness variant of the same one. In practice, $f_0$ is typically determined by domain convention. Our use of the general notation $f_0$ reflects this range of applications rather than an arbitrary modeling choice.

## D. Verification of Assumptions for the Examples

In this section, we verify that the examples introduced in Section 2 satisfy Assumptions 3.2 and 3.3 under standard regularity conditions.

For linear regression with squared loss, Assumption 3.2 requires compact convex $\Theta$ and finite second moments; all properties hold when $\Sigma_g$ is positive definite, while only (1)-(2) hold in overparameterized settings. For logistic regression with binary cross-entropy, Assumption 3.2 requires compact convex $\Theta$, finite first moments, and $X \neq 0$ with positive probability. For RKHS methods with squared loss, Assumption 3.3 requires $\mathbf{E}_g[k(X, X)] < \infty$. The nonparametric assumption also encompasses Lipschitz/Hölder functions (via Arzelà-Ascoli) and sieves/basis expansions.

**Linear Regression.** Consider a function class $\mathcal{F} = \{f(x) = \theta^T x : \theta \in \Theta\}$ with a convex and compact set $\Theta$ and squared loss $\ell(y, f(x)) = (y - f(x))^2$. The data generation process is $Y = \beta_g^T X + \epsilon_g$ with $\mathbf{E}_g[\epsilon_g | X] = 0$, $\epsilon_g \sim \mathcal{N}(0, \sigma_g^2)$ and $\mathbf{E}_g[XX^T] = \Sigma_g$.

We verify that Assumption 3.2 holds:

**(1) Strict convexity and continuity.** The squared loss $\ell(y, \theta^\top x) = (y - \theta^\top x)^2$ is continuous with respect to $\theta$. For strict convexity, note that

$$R_g(\theta) = \mathbf{E}_g[(Y - \theta^T X)^2] = \|\theta - \beta_g\|_{\Sigma_g}^2 + \sigma_g^2,$$

where $\|\theta\|_{\Sigma_g}^2 = \theta^T \Sigma_g \theta$. If $\Sigma_g$ is positive definite, then $R_g(\theta)$ is strictly convex in $\theta$. In the overparameterized case where $\Sigma_g$ is not positive definite, the risk is convex but not strictly convex: if $v \in \ker(\Sigma_g)$, then $R_g(\theta + tv) = R_g(\theta)$ for all $t \in \mathbb{R}$.

**(2) Dominating function.** For any $\theta \in \Theta$,

$$\begin{aligned}
\ell(y, f_\theta(x)) = (y - \theta^\top x)^2 &\leq 2y^2 + 2(\theta^\top x)^2 \\
&\leq 2y^2 + 2\|\theta\|^2 \|x\|^2 \\
&\leq 2y^2 + 2B^2 \|x\|^2 =: h(x, y),
\end{aligned}$$

where $B = \sup_{\theta \in \Theta} \|\theta\| < \infty$ by compactness of $\Theta$. Since $\mathbf{E}_g[Y^2] = \mathbf{E}_g[(\beta_g^T X + \epsilon_g)^2] < \infty$ (as $\mathbf{E}_g[\|X\|^2] < \infty$ and $\sigma_g^2 < \infty$) and $\mathbf{E}_g[\|X\|^2] < \infty$, we have $\mathbf{E}_g[h(X, Y)] < \infty$ for all $g$.

**(3) Compact and convex parameter space.** The parameter set $\Theta$ is compact and convex by assumption.

Therefore, all conditions of Assumption 3.2 are satisfied for the linear regression problem with finite second moment of $\|X\|$.

**Logistic Regression.** Consider a function class $\mathcal{F} = \{f(x) = \sigma(\theta^T x) : \theta \in \Theta\}$, where $\sigma(\cdot)$ denotes the sigmoid function and $\Theta$ denotes a convex and compact parameter set. Let the loss function be the binary cross-entropy $\ell(y, f(x)) = -y \log f(x) - (1 - y) \log(1 - f(x))$ for $y \in \{0, 1\}$. The data generation process is $Y = \mathbf{1}\{\beta_g^T X + \epsilon > 0\}$ where $X \sim N(0, \Sigma_g)$ and $\epsilon \sim N(0, \sigma_g^2)$.

We verify that Assumption 3.2 holds:

**(1) Convexity and continuity.** The binary cross-entropy loss can be written as

$$\ell(y, f_\theta(x)) = -y \log \sigma(\theta^\top x) - (1 - y) \log(1 - \sigma(\theta^\top x)) = \log(1 + e^{\theta^\top x}) - y\theta^T x.$$

This function is convex in $\theta$, and the risk function $R_g(\theta) = \mathbf{E}_g[\ell(Y, f_\theta(X))]$ is convex. Continuity with respect to $\theta$ is immediate from continuity of the exponential and logarithm functions.

**(2) Dominating function.** The loss function satisfies

$$\begin{aligned}
|\ell(y, f_\theta(x))| = |\log(1 + e^{\theta^T x}) - y\theta^\top x| &\leq \log(1 + e^{|\theta^\top x|}) + |\theta^T x| \\
&\leq \log 2 + 2|\theta^T x| \\
&\leq \log 2 + 2B\|x\| =: h(x, y),
\end{aligned}$$

where we used $\log(1 + e^t) \leq \log 2 + |t|$ for all $t \in \mathbb{R}$, and $B = \sup_{\theta \in \Theta} \|\theta\| < \infty$ by compactness. Since $\mathbf{E}_g[\|X\|] < \infty$ by assumption, we have $\mathbf{E}_g[h(X, Y)] < \infty$ for all $g$.

**(3) Compact and convex parameter space.** The parameter set $\Theta$ is compact and convex by assumption.

Therefore, all conditions of Assumption 3.2 are satisfied for the logistic regression problem with binary cross-entropy and finite first moment of $\|X\|$.

**RKHS Setting.** For a positive semi-definite kernel $k : \mathcal{X} \times \mathcal{X} \to \mathbf{R}$ defining an RKHS $\mathcal{H}$, consider the function class

$$\mathcal{F} = \{f \in \mathcal{H} : \|f\|_{\mathcal{H}} \leq R\}$$

with squared loss $\ell(y, t) = (y - t)^2$.

We verify that Assumption 3.3 holds:

**(1) Strict convexity and continuity.** The squared loss $\ell(y, t) = (y - t)^2$ is strictly convex and continuous with respect to $t$.

**(2) Dominating function.** Since $\mathcal{H}$ is an RKHS, every $f \in \mathcal{F}$ satisfies

$$|f(x)| \leq \|f\|_{\mathcal{H}} \sqrt{k(x,x)} \leq R\sqrt{k(x,x)}.$$

Therefore,

$$|\ell(y, f(x))| \leq 2y^2 + 2|f(x)|^2 \leq 2y^2 + 2R^2 k(x,x) =: g(x,y).$$

If $\mathbf{E}_g[Y^2] < \infty$ and $\mathbf{E}_g[k(X,X)] < \infty$ for all $g$, then $\mathbf{E}_g[g(X,Y)] < \infty$ for all $g$.

**(3) Weak compactness.** The set $\mathcal{F} = \{f \in \mathcal{H} : \|f\|_{\mathcal{H}} \leq R\}$ is the closed ball of radius $R$ in the Hilbert space $\mathcal{H}$. By the Banach-Alaoglu theorem (or equivalently, the weak compactness of closed balls in Hilbert spaces), $\mathcal{F}$ is weakly compact. Moreover, $\mathcal{F}$ is convex by definition. Evaluation maps $f \mapsto f(x)$ are continuous with respect to the weak topology since $f(x) = \langle f, k(\cdot, x)\rangle_{\mathcal{H}}$.

Therefore, all conditions of Assumption 3.3 are satisfied for the RKHS setting with squared loss when $\mathbf{E}_g[Y^2] < \infty$ and $\mathbf{E}_g[k(X,X)] < \infty$ for all groups $g$. We provide examples of nonparametric function classes that satisfy the assumptions under standard regularity conditions.

**Lipschitz and Hölder Functions.** For $\mathcal{F} = \{f : |f(x) - f(x')| \leq L\|x - x'\|^{\alpha}, \|f\|_{\infty} \leq M\}$ with $\mathcal{X}$ compact, the function class is equicontinuous and uniformly bounded. By the Arzelà-Ascoli theorem, $\mathcal{F}$ is compact under the sup-norm topology. Evaluation maps $f \mapsto f(x)$ are continuous since $\|f_n - f\|_{\infty} \to 0$ implies $|f_n(x) - f(x)| \to 0$ for all $x$. Convexity is immediate: if $f_1, f_2 \in \mathcal{F}$ and $0 < \lambda < 1$, then

$$|(\lambda f_1 + (1-\lambda)f_2)(x) - (\lambda f_1 + (1-\lambda)f_2)(x')| \leq \lambda L\|x - x'\|^{\alpha} + (1-\lambda)L\|x - x'\|^{\alpha} = L\|x - x'\|^{\alpha},$$

and $\|\lambda f_1 + (1-\lambda)f_2\|_{\infty} \leq \lambda M + (1-\lambda)M = M$.

**Sieves and Basis Expansions.** For $\mathcal{F}_k = \{\sum_{j=1}^{k} \theta_j \phi_j : \theta \in \Theta\}$ where $\Theta \subset \mathbb{R}^k$ is compact and convex and $\{\phi_j\}$ are continuous basis functions (e.g., wavelets, splines), the map $R : \Theta \to C(\mathcal{X})$ defined by $R(\theta)(x) = \sum_{j=1}^{k} \theta_j \phi_j(x)$ is continuous in the sup-norm topology. Since $\Theta$ is compact, $\mathcal{F}_k = R(\Theta)$ is compact. Evaluation maps are continuous: if $\theta_n \to \theta$ in $\Theta$, then $\sum_{j=1}^{k} \theta_{n,j} \phi_j(x) \to \sum_{j=1}^{k} \theta_j \phi_j(x)$ for all $x$. Convexity follows from convexity of $\Theta$: if $f_1 = \sum_j \theta_j^{(1)} \phi_j$ and $f_2 = \sum_j \theta_j^{(2)} \phi_j$, then $\lambda f_1 + (1-\lambda)f_2 = \sum_j (\lambda \theta_j^{(1)} + (1-\lambda)\theta_j^{(2)})\phi_j \in \mathcal{F}_k$.

# E. Discussion of the Bargaining Problem

## E.1. Axioms for Other Bargaining Solutions

We provide formal definitions of axioms referenced in Section 4.4. The four core axioms (PO), (SYM), (SI), and (IM) are defined in the main text. Here we define additional axioms satisfied by alternative bargaining solutions.

1. **Weak Pareto Optimality (WPO).** There exists no $f' \in \mathcal{F}$ such that $R_g(f') < R_g(f)$ for all $g \in \mathcal{G}$. This is weaker than (PO), which requires that no alternative weakly improves all components and strictly improves at least one.

2. **Independence of Irrelevant Alternatives (IIA).** If $\mathcal{F}_1 \subseteq \mathcal{F}_2$ with the same baseline $f_0$ and disagreement point, and the solution under $\mathcal{F}_2$ satisfies $f_2 \in \mathcal{F}_1$, then $f_1 = f_2$.

3. **Translation Invariance (TI).** For any constants $\{c_g\}_{g=1}^{m}$, the affine transformation $\widetilde{R}_g(f) = R_g(f) + c_g$ preserves the solution structure: $\widetilde{R}_g(\widetilde{f}) = R_g(f) + c_g$ for all $g$, where $\widetilde{f}$ denotes the solution under the transformed problem.

4. **Strong Monotonicity (SM).** If $\mathcal{F}_1 \subseteq \mathcal{F}_2$ with the same baseline $f_0$, then $R_g(f_1) \geq R_g(f_2)$ for all $g \in \mathcal{G}$, where $f_1$ and $f_2$ denote the solutions under $\mathcal{F}_1$ and $\mathcal{F}_2$, respectively.

5. **Strong Monotonicity other than Ideal Point (SMON).** If $\mathcal{F}_1 \subseteq \mathcal{F}_2$ with the same group-optimal risks $R_g(f_g^*)$ for all $g$, then $R_g(f_1) \geq R_g(f_2)$ for all $g \in \mathcal{G}$, where $f_1$ and $f_2$ denote the solutions under $\mathcal{F}_1$ and $\mathcal{F}_2$, respectively.

**Nash Bargaining Solution** The Nash bargaining solution (Nash et al., 1950) maximizes the product of utility gains: $\max_{u \in S} \prod_{i=1}^{m}(u_i - d_i)$. It is uniquely characterized by (PO), (SYM), (SI), and (IIA).

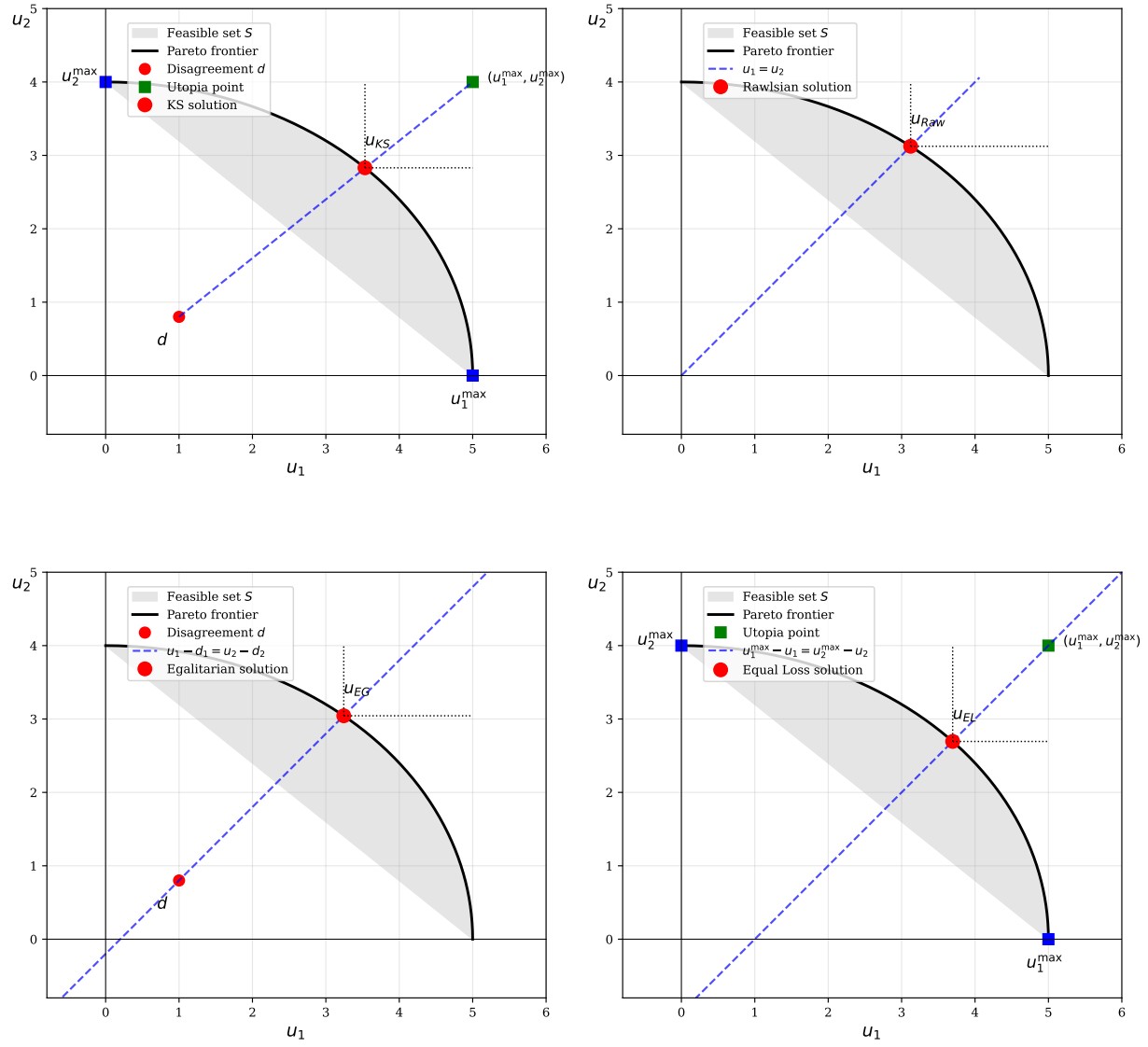

*Figure 7.* Comparison of bargaining solutions: (a) KS equalizes relative improvements, (b) Rawlsian maximizes minimum utility, (c) Egalitarian equalizes absolute gains from $d$, (d) Equal Loss equalizes regrets from ideal points.

**Egalitarian Solution** The egalitarian solution (Kalai, 1977) selects the outcome where all players achieve equal gain from the disagreement point: $u_i - d_i = u_j - d_j$ for all $i, j$. It satisfies (SYM), (TI), and (SM), but only guarantees weak Pareto optimality in general. Chen (2000) showed that in two-player games, when the egalitarian solution coincides with the maximin gain solution, full (PO) is guaranteed.

**Equal Loss Solution** The equal loss solution (Chun, 1988) selects outcomes where all players suffer equal loss from their ideal points: $b_i - u_i = b_j - u_j$ for all $i, j$, where $b_i$ is player $i$'s maximum achievable utility. Like the egalitarian solution, it satisfies (WPO), (SYM), (TI), and (SMON), where (SMON) is a slight modification of (SM) with respect to the ideal point rather than the disagreement point.

See Figure 7 for illustration of each bargaining solution in two player setting.

## E.2. Leximin Refinement and Comprehensive Closure

### E.2.1. LEXIMIN REFINEMENT

**Definition E.1** (Leximin solution). A leximin solution is defined through a sequential maximization process. For a predictor $f \in \mathcal{F}$, let $\boldsymbol{\rho}(f) = (\rho_1(f), \cdots, \rho_m(f))$ denote its relative improvement vector, and let $\rho_{(1)}(f) \leq \cdots \leq \rho_{(m)}(f)$ denote the sorted components. The leximin solution $f_{\text{RI}}^*$ is found by:

1. First, maximize the minimum relative improvement:
$$\mathcal{F}_1 = \arg\max_{f \in \mathcal{F}} \min_{g \in \mathcal{G}} \rho_g(f) = \arg\max_{f \in \mathcal{F}} \rho_{(1)}(f).$$

2. Among predictors in $\mathcal{F}_1$, maximize the second-smallest relative improvement:
$$\mathcal{F}_2 = \arg\max_{f \in \mathcal{F}_1} \rho_{(2)}(f)$$

3. Continue sequentially: for $k = 3, \ldots, m$,
$$\mathcal{F}_k = \arg\max_{f \in \mathcal{F}_{k-1}} \rho_{(k)}(f)$$

4. The set of leximin solutions is $\mathcal{F}_m$.

Equivalently, we can write this as a single lexicographic maximization:
$$f_{\text{RI}}^* = \arg\max_{f \in \mathcal{F}}^{\text{lex}} \left( \rho_{(1)}(f), \ldots, \rho_{(m)}(f) \right) \tag{28}$$

*Remark* E.2 (Tie-breaking and uniqueness). At each stage $k$, if $\mathcal{F}_k$ contains multiple predictors, the next stage breaks ties by maximizing $\rho_{(k+1)}$. Under the regularity conditions of Theorem 3.4, the sequential process terminates at a unique relative improvement vector and it is Pareto optimal. This is the result of Imai (1983).

*Remark* E.3 (Connection to maximin). The first stage $\mathcal{F}_1$ corresponds to the maximin solution that maximizes worst-case relative improvement. The leximin refinement provides a principled way to break ties when multiple predictors achieve the same worst-case performance, by prioritizing improvements to successively less-advantaged groups.

*Remark* E.4 (Leximin refinement for other bargaining solutions). The leximin refinement is not unique to the Kalai-Smorodinsky solution. As mentioned in the main text (footnote 2), it can be applied to resolve non-uniqueness in other bargaining solutions for $m > 2$ groups. The choice of which vector to leximin-optimize (relative improvements for KS, utilities for Egalitarian) depends on the underlying fairness criterion being refined.

### E.2.2. COMPREHENSIVE CLOSURE

Classical bargaining theory employs different notions of comprehensiveness depending on the role of the disagreement point. In the main text, we adopt $d$-comprehensiveness as the comprehensive condition. The terminology below follows the definitions from Thomson (1994).

**Definition E.5** (Comprehensive set). A set $S \subseteq \mathbb{R}^m$ is *comprehensive* if whenever $\boldsymbol{x} \in S$ and $\boldsymbol{y} \leq \boldsymbol{x}$ componentwise (i.e., $y_i \leq x_i$ for all $i$), then $\boldsymbol{y} \in S$. Comprehensiveness allows utility to be disposed of in any amount without bound.

**Definition E.6** ($d$-Comprehensive set). Given a disagreement point $\boldsymbol{d} \in \mathbb{R}^m$, a set $S \subseteq \mathbb{R}^m$ is $\boldsymbol{d}$-*comprehensive* if whenever $\boldsymbol{x} \in S$ and $\boldsymbol{d} \leq \boldsymbol{y} \leq \boldsymbol{x}$ componentwise, then $\boldsymbol{y} \in S$. This property captures the assumption that utility is freely disposable above the disagreement point $\boldsymbol{d}$.

*Remark* E.7 (Relationship between notions). A $\boldsymbol{d}$-comprehensive set allows voluntary utility disposal only down to the disagreement point $\boldsymbol{d}$, reflecting the assumption that rational players would not voluntarily accept utilities below what they are guaranteed at disagreement (Individual Rationality, as discussed in Theorem 4.1).

*Remark* E.8 (When $f_0 \notin \mathcal{F}$). Theorem 4.1 assumes $f_0 \in \mathcal{F}$ to guarantee individual rationality. When $f_0 \notin \mathcal{F}$, some groups may have $\rho_g(f_{\text{RI}}) < 0$, meaning worse performance than baseline. However, the bargaining problem remains well-defined.

Following Thomson (1994), a bargaining problem $(S, \boldsymbol{d})$ is non-degenerate if:
$$\exists x \in S \text{ such that } x_i > d_i \text{ for all } i \in \{1, \ldots, m\}. \tag{29}$$

In our framework, this translates to:

$$\exists f \in \mathcal{F} \text{ such that } R_g(f) < R_g(f_0) \text{ for all } g \in \mathcal{G}. \tag{30}$$

Without this condition, every predictor in $\mathcal{F}$ harms at least one group relative to baseline, making the bargaining problem degenerate. The assumption $f_0 \in \mathcal{F}$ in Theorem 4.1 is a sufficient (but not necessary) condition that ensures both non-degeneracy and individual rationality. When $f_0 \notin \mathcal{F}$, condition (30) ensures non-degeneracy and individual rationality.

In degenerate cases where condition (30) fails, one could technically use the fully comprehensive closure instead of the $d$-comprehensive closure to keep the solution well-defined. However, this would permit $\rho_g < 0$, violating the fairness principle that cooperation should not harm participants.

**Definition E.9** ($d$-Comprehensive closure)**.** The $d$-comprehensive closure of $S$ with respect to disagreement point $d$ is

$$\text{comp}_{\boldsymbol{d}}(S) = \{\boldsymbol{y} \in \mathbb{R}^m : \exists \boldsymbol{x} \in S \text{ such that } \boldsymbol{d} \le \boldsymbol{y} \le \boldsymbol{x} \text{ componentwise}\}.$$

This is the smallest $\boldsymbol{d}$-comprehensive set containing $S$.

*Remark* E.10 (Application to our framework)*.* In relative improvement space, the disagreement point is $\boldsymbol{d} = \boldsymbol{0}$ (corresponding to the baseline predictor). The $\boldsymbol{0}$-comprehensive closure is

$$\text{comp}_{\boldsymbol{0}}(\Omega(\mathcal{F})) = \{\boldsymbol{\rho}' \in \mathbb{R}^m : \exists \boldsymbol{\rho} \in \Omega(\mathcal{F}) \text{ such that } \boldsymbol{0} \le \boldsymbol{\rho}' \le \boldsymbol{\rho} \text{ componentwise}\}.$$

# F. Figure Descriptions

## F.1. Detailed Explanation of Motivating Example

Figure 1 (and Figure 9) provides an alternative view of the Pareto frontier by reparametrizing it through the slope coefficient $\beta_1$ of the single-feature linear model $f(x) = \beta_1 x + \beta_0$ (see Appendix G for the Pareto frontier construction).

The *left panel* of each subfigure plots per-group RMSE $R_g(\beta_1)$ as a function of $\beta_1$. Horizontal dotted lines indicate the group baselines $R_g(f_0)$, and filled circles mark the group-specific oracle slopes $\beta_{1,g}^*$ at which each group's risk is individually minimized. The *right panel* plots relative improvement $\rho_g(\beta_1) = (R_g(f_0) - R_g(\beta_1)) / (R_g(f_0) - R_g(f_g^*))$ as a function of $\beta_1$. Vertical dashed lines indicate the $\beta_1$ selected by MMR (purple) and MMRI (green), with the achieved RI values annotated at each solution. The divergence between the two $\rho_g$ curves at the MMR solution directly reflects the asymmetry in oracle gaps between groups.

## F.2. Detailed Comparison of Group Fairness Methods

Figure 3 illustrates how common group fairness criteria select different solutions along the Pareto frontier in risk space, corresponding to the objectives in Equation (3), (12)–(14).

Maximin relative improvement selects the Pareto-optimal solution that maximizes the minimum relative improvement across groups. As discussed in Section 3.2, for the two-group case ($m = 2$), this solution coincides with the equal relative improvement point on the Pareto frontier. Consequently, it is given by the intersection of the Pareto frontier with the equal-relative-improvement line, i.e., the line segment connecting the disagreement point and the utopia point.

Group DRO minimizes the worst-group risk, selecting the Pareto-optimal point that equalizes the maximum group risk. As noted by Martinez et al. (2020), an equal-risk point may not exist; in this case, it selects the point on the Pareto frontier that is closest to equal risk across groups, which is the minimax Pareto-fair solution.

Maximin explained variance (MMV) maximizes the minimum absolute improvement from the baseline across groups; geometrically, it corresponds to the point on the Pareto frontier that is closest to the line of slope one passing through the disagreement point, thereby balancing absolute risk reductions across groups.

Minimax regret (MMR) minimizes the maximum deviation from each group's optimal risk; geometrically, it corresponds to the point on the Pareto frontier that intersects the equal-regret line of slope one emanating from the utopia (ideal) point.

However, when the feasible set is rectangular, the Pareto frontier collapses to a single point. In this case, the equal-regret line does not intersect the Pareto frontier, and the unique Pareto-optimal point—coinciding with the utopia point—becomes the solution for all methods. In this case, the solution is neither an equal-risk nor an equal-regret point; nevertheless, it still satisfies equal relative improvement.

### F.3. Linear regression setups

Figure 4 presents synthetic data analyses using linear regression with two groups. We consider the linear regression setup described in Section 2. We use the function class $\mathcal{F} = \{f(x) = \theta^\top x : \theta \in \Theta\}$ with convex compact $\Theta$ and squared loss. We assume that $Y = \beta_g^\top X + \epsilon_g$ with $\mathbf{E}_g[\epsilon_g|X] = 0$, $\epsilon_g \sim \mathcal{N}(0, \sigma_g^2)$, $\mathbf{E}_g[X] = 0$ and $\mathbf{E}_g[XX^\top] = \Sigma_g$. We use the natural baseline in regression which is $f_0(x) = 0$ (the unconditional mean) and under the squared loss, the group optimum is $f_g^*(x) = \beta_g^\top x$. Then the optimization reduces to

$$\theta_{\text{RI}} = \arg\max_{\theta \in \Theta} \min_{g \in \mathcal{G}} \left( 1 - \frac{\|\theta - \beta_g\|_{\Sigma_g}^2}{\|\beta_g\|_{\Sigma_g}^2} \right).$$

In Figure 4, we illustrate the multiple linear regression example with two predictors:

$$\Theta = \{\theta : \|\theta\| \leq 1\}, \quad \beta_1 = (0.4, 0), \quad \beta_2 = (0.4, 0.6),$$
$$\Sigma_1 = \left( \begin{smallmatrix} 1 & 0.5 \\ 0.5 & 1 \end{smallmatrix} \right), \quad \Sigma_2 = \left( \begin{smallmatrix} 1 & 0 \\ 0 & 1 \end{smallmatrix} \right), \quad \sigma_g = 1.$$

Rather than specifying solutions, we focus on the geometric properties of the risk set $\mathcal{R}(\mathcal{F})$. Figure 4 illustrates that $\mathcal{U}$ is compact and convex, and shows its convex and comprehensive hull. As seen in the figure, the Pareto frontier remains unchanged after taking the hull, so the bargaining solutions of interest are unaffected by this operation.

## G. Empirical Illustration on ACS Income data

**Data and setup.** We use the American Community Survey (ACS) public-use microdata for 2018, accessed via the *folktables* package (Ding et al., 2021). The prediction target is log-transformed total personal income (PINCP), restricted to full-time, year-round employees (ESR = 1, WKHP $\geq$ 35, WKW = 1). We consider two binary group partitions—*sex* (male/female) and *race* (white/non-white)—across 50 U.S. states.

The feature set comprises four standard ACS predictors: age (AGEP), educational attainment (SCHL), and two binary indicators derived from categorical ACS variables: marital status MARC = $\mathbf{1}\{$MAR = $1\}$ (currently married vs. not), and householder status RELPC = $\mathbf{1}\{$RELP = $0\}$ (reference person of the household vs. not).

**Implementation.** The model class $\mathcal{F}$ is linear regression with intercept, parametrized by group-reweighting $\lambda \in [0, 1]$:

$$\min_{\beta, \beta_0} (1 - \lambda) \frac{1}{n_0} \sum_{i: g_i = 0} (y_i - f_\beta(x_i))^2 + \lambda \frac{1}{n_1} \sum_{i: g_i = 1} (y_i - f_\beta(x_i))^2, \tag{31}$$

admitting the closed-form solution $\hat{\beta} = (X^\top W X)^{-1} X^\top W y$. The baseline is the constant predictor $f_0(x) = \bar{y}$, and per-group oracles $f_g^*$ are fit separately on each group. Sweeping $\lambda$ over 10,001 equally spaced values traces the full Pareto frontier. Since $\mathcal{R}(\mathcal{F})$ is compact and its convex hull contains no Pareto optimal points outside $\mathcal{R}(\mathcal{F})$ (Theorem 3.4), sweeping $\lambda \in [0, 1]$ over the scalarized objective (31) traces the complete Pareto frontier.

Each fairness criterion selects its solution from this frontier: *ERM* uses sample-proportion weighting; *Group DRO* minimizes $\max_g \hat{R}_g(f)$; *MMR* minimizes $\max_g(\hat{R}_g(f) - \hat{R}_g(f_g^*))$; *MMV* maximizes $\min_g(\hat{R}_g(f_0) - \hat{R}_g(f))$; *MMRI* maximizes $\min_g \hat{\rho}_g$, where $\hat{\rho}_g = (\hat{R}_g(f_0) - \hat{R}_g) / (\hat{R}_g(f_0) - \hat{R}_g(f_g^*))$; *Nash* maximizes $\prod_g (\hat{R}_g(f_0) - \hat{R}_g(f))$.

For nonlinear models, we implement each method via iterative optimization using a two-phase procedure: an ERM warm-start for $T_{\text{warm}}$ epochs, followed by method-specific exponentiated gradient ascent on group weights $q_g$. *Group DRO* updates $q_g \propto q_g \exp(\eta \hat{R}_g^2)$; *MMR* replaces losses with regrets; *MMRI* updates $q_g \propto q_g \exp(-\eta \hat{\rho}_g)$, up-weighting the group with the lowest relative improvement. Results from the gradient-based procedure are consistent with the closed-form solutions across all configurations considered.

**Gap ratio distribution.** Table 2 summarizes the oracle gap ratio $r = (R_0(f_0) - R_0(f_0^*))/(R_1(f_0) - R_1(f_1^*))$ across all 400 configurations (50 states $\times$ 2 partitions $\times$ 4 features). Asymmetry is considerably more pronounced under the race partition than the sex partition: for householder status (RELPC), 86% of states yield extreme ratios ($r < 0.5$ or $r > 2.0$), and no state is near-symmetric. Under the sex partition, marital status (MARC) shows the most asymmetry (14% extreme), while education (SCHL) is the most symmetric (66% near-symmetric).

*Table 2.* **Oracle gap ratio distribution across 50 U.S. states.** For each partition–feature combination, we report the min, median, and max of the gap ratio $r$ across states, along with the fraction of states with extreme asymmetry ($r < 0.5$ or $r > 2.0$) and near-symmetry ($0.8 \leq r \leq 1.25$).

| Partition | Feature | Min | Median | Max | Extreme | Symmetric |
|---|---|---|---|---|---|---|
| sex | AGEP | 0.58 | 1.32 | 2.02 | 1/50 (2%) | 19/50 (38%) |
| | SCHL | 0.33 | 0.88 | 1.31 | 2/50 (4%) | 33/50 (66%) |
| | MARC | 0.71 | 1.62 | 2.18 | 7/50 (14%) | 4/50 (8%) |
| | RELPC | 0.37 | 1.01 | 1.81 | 3/50 (6%) | 26/50 (52%) |
| race | AGEP | 0.10 | 0.61 | 3.48 | 17/50 (34%) | 10/50 (20%) |
| | SCHL | 0.15 | 0.69 | 1.84 | 6/50 (12%) | 12/50 (24%) |
| | MARC | 0.18 | 0.53 | 2.96 | 25/50 (50%) | 4/50 (8%) |
| | RELPC | 0.04 | 0.30 | 2.85 | 43/50 (86%) | 0/50 (0%) |

**Results.** We first examine single-feature models across three states—California, Hawaii, and North Dakota—chosen because they exhibit pronounced gap asymmetry in different directions and magnitudes. Figure 8 shows the Pareto frontier and method solutions for six configurations; Figure 9 reparametrizes the same frontiers by the slope coefficient $\beta$, displaying per-group risk and relative improvement as functions of $\beta$.

*California, sex, marital status (*`MARC`*).* The marriage wage premium is well documented for men but substantially weaker for women (Korenman & Neumark, 1991). This asymmetry appears directly in the oracle gaps: marital status reduces prediction error nearly twice as much for men as for women (gap ratio $\approx 1.90$). MMR over-allocates to the male group (Figure 9a).

*California, race, age (*`AGEP`*).* Age predicts income roughly twice as well for white workers as for non-white workers (gap ratio $\approx 2.14$), consistent with differential returns to experience across racial groups (Figure 9b).

*Hawaii, sex, marital status (*`MARC`*).* Hawaii exhibits a similar marriage premium asymmetry (gap ratio $\approx 2.17$), with a distinctive demographic composition that amplifies the effect (Figure 9c).

*Hawaii, race, age (*`AGEP`*).* The gap ratio reaches $\approx 3.48$—the most extreme among all 50 states—reflecting the unique racial composition of Hawaii's labor market, where age predicts white workers' income far more strongly than non-white workers' income. At the MMR solution, the non-white group attains only $24\%$ of its oracle improvement while the white group attains $78\%$ (Figure 9d).

*North Dakota, sex, education (*`SCHL`*).* Education reduces prediction error roughly three times more for women than for men (gap ratio $\approx 0.33$). At the MMR solution, the female group attains $58\%$ of its oracle improvement while the male group—with three times less room to improve—is made worse than baseline ($-28\%$). MMRI equalizes relative improvement across groups (Figure 9e).

*North Dakota, race, age (*`AGEP`*).* Age is roughly ten times more predictive for non-white workers than for white workers (gap ratio $\approx 0.10$). MMR allocates nearly all model capacity to the non-white group, leaving the white group worse than baseline—an extreme instance of scale insensitivity (Figure 9f).

In all six cases, MMRI equalizes relative improvement across groups, while MMR systematically over-serves the group whose oracle gap is larger in absolute terms.

The asymmetry persists under the full four-feature model (Figure 10). In California, additional features compress the ratio toward symmetry—1.22 (sex) and 1.14 (race). In Hawaii, the race partition retains a notable gap ratio of 1.71 even with all four features. In North Dakota, the gap ratio remains substantially below unity: 0.53 (sex) and 0.32 (race). Even moderate asymmetry is sufficient for MMR to allocate disproportionately across groups. Taken together, these results confirm that the failure mode illustrated in Figure 1 is not an artifact of the synthetic construction, but a systematic consequence of applying absolute-scale criteria when groups differ in inherent predictability.

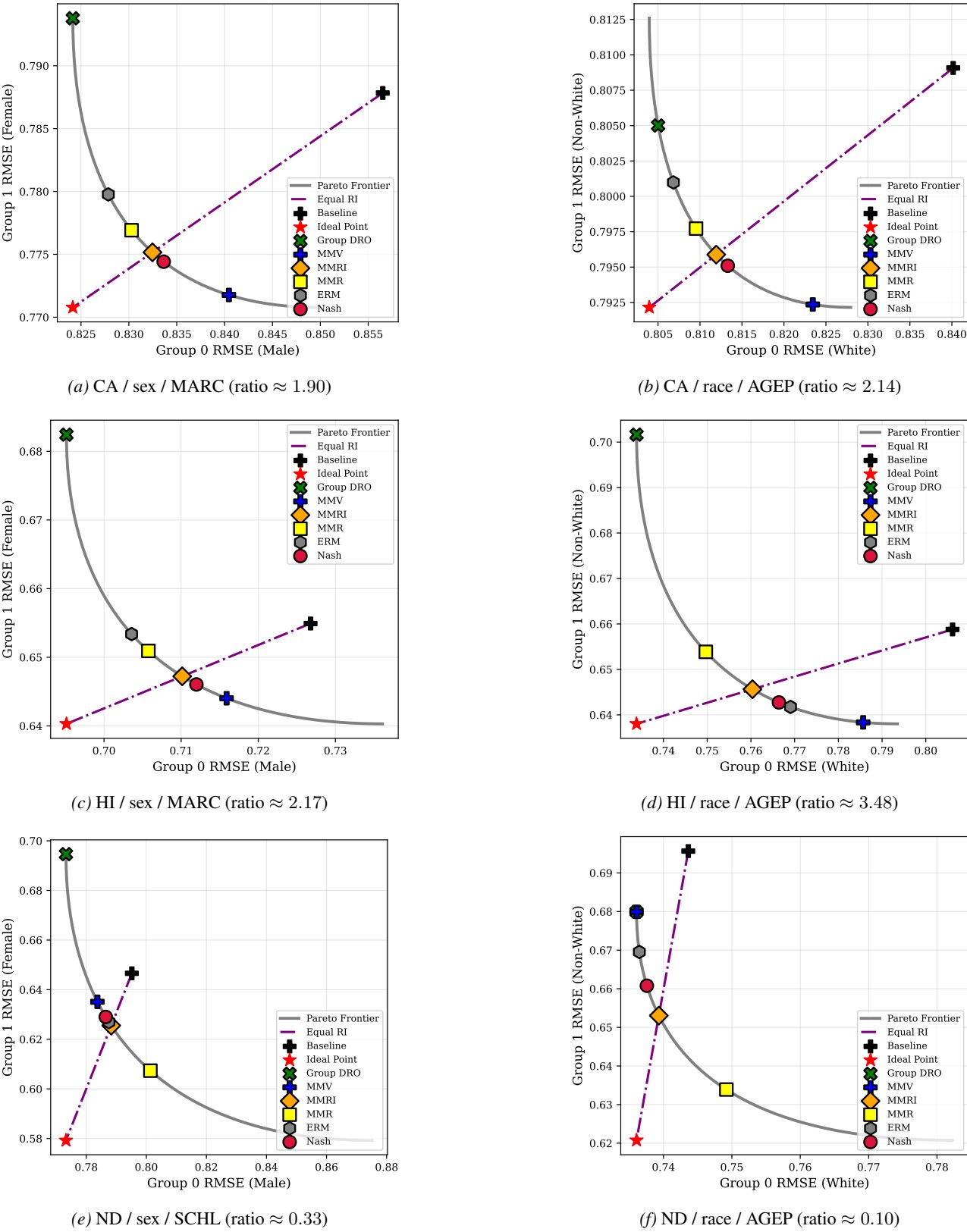

*Figure 8.* **Pareto frontiers and method solutions (single-feature models).** Each panel shows per-group RMSE for the Pareto-optimal set of linear models (grey curve), the baseline (cross), the ideal point (star), and the solutions selected by ERM, Group DRO, MMR, MMRI, MMV, and Nash. The dashed line indicates equal relative improvement. When the gap ratio deviates from unity, MMR moves away from the equal-RI line, while MMRI remains on or near it.

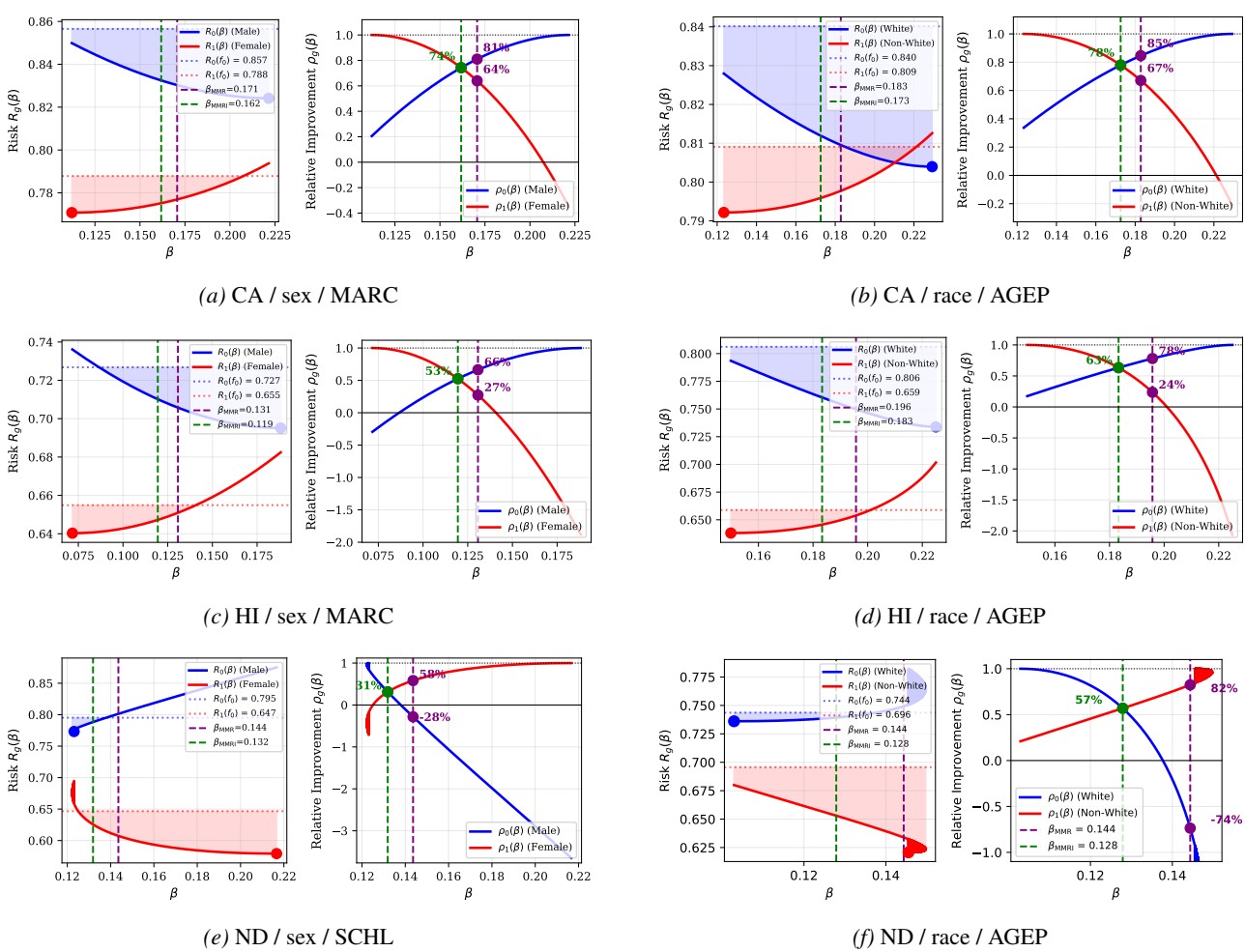

*Figure 9.* **Risk and relative improvement as functions of the slope $\beta$ (single-feature models).** Left half of each panel: per-group RMSE $R_g(\beta)$ with baselines (dotted) and oracle points (circles). Right half: relative improvement $\rho_g(\beta)$, with MMR (purple) and MMRI (green) solutions marked. The gap between the two curves at each $\beta$ reflects the asymmetry in oracle gaps.

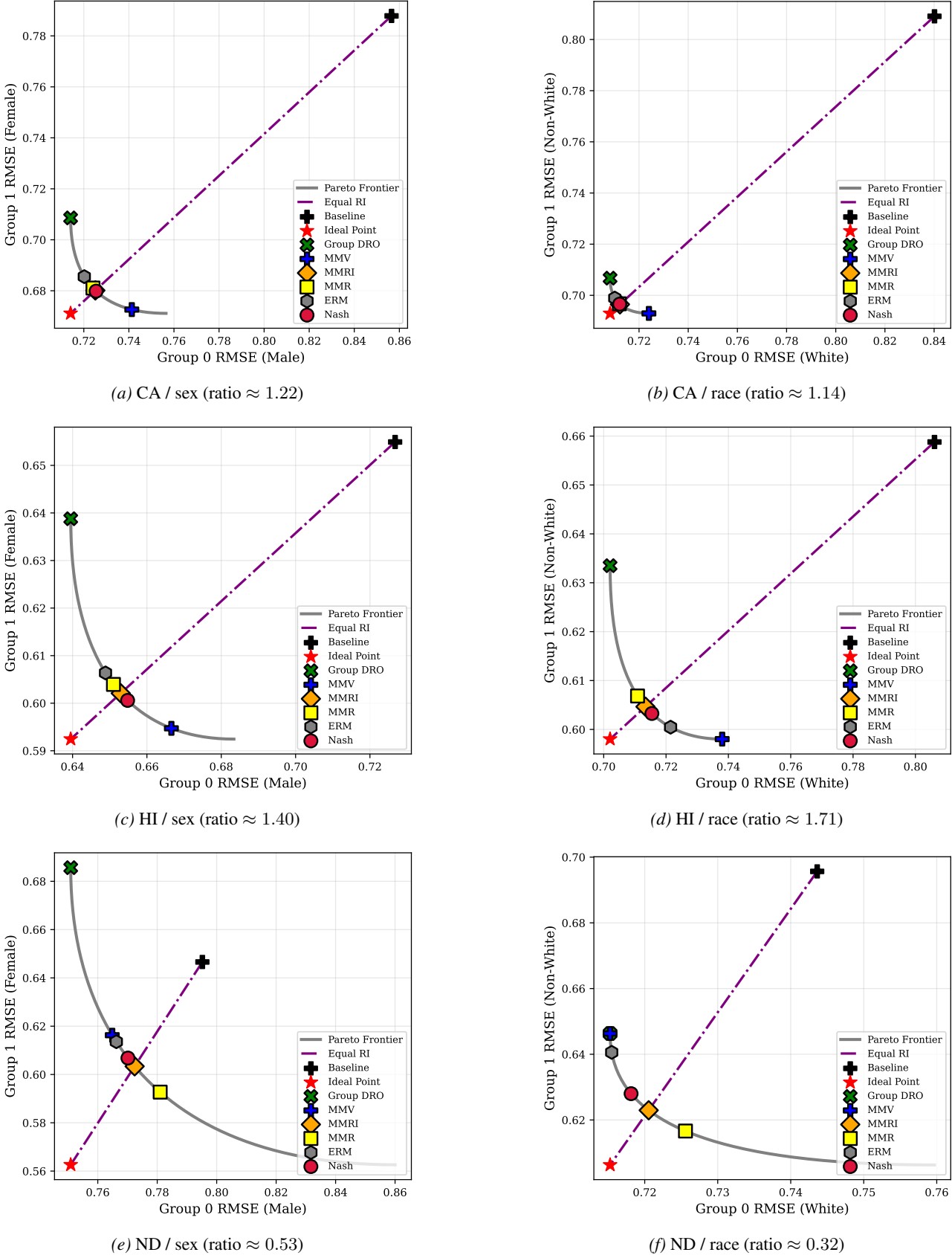

*(a)* CA / sex (ratio ≈ 1.22)

*(b)* CA / race (ratio ≈ 1.14)

*(c)* HI / sex (ratio ≈ 1.40)

*(d)* HI / race (ratio ≈ 1.71)

*(e)* ND / sex (ratio ≈ 0.53)

*(f)* ND / race (ratio ≈ 0.32)

*Figure 10.* **Pareto frontiers and method solutions (full four-feature models).** Same layout as Figure 8. The gap asymmetry narrows relative to single-feature models but remains present, particularly for North Dakota under the race partition (ratio 0.32) and Hawaii under the race partition (ratio 1.71).

