# OpenReview forum: "Maximin Relative Improvement: Fair Learning as a Bargaining Problem"
_ICML.cc/2026/Conference — ICML 2026 regular_

### Official Review · Reviewer_tqDM · 2026-02-17

**Soundness:** 3
**Presentation:** 3
**Significance:** 2
**Originality:** 2
**Overall Recommendation:** 3
**Confidence:** 3

**Summary:**

This paper interprets fairness in machine learning as a bargaining problem, and proposes a new fairness objective: maximize the minimum relative improvement across groups, where relative improvement is the ratio of improvement of error (relative to a baseline) on the group compared to the optimal improvement possible over the baseline with a group specific classifier in some class F. They then note that solutions to this problem correspond to the Kalai–Smorodinsky solution, which endows them with a number of nice axiomatic properties (only for convex classes) when there are two groups. They note an extension in the bargaining literature to m > 2 groups, but this requires solving for a leximin optimal point rather than just a maximin optimal point, and the authors leave open both algorithmic and sample complexity issues in this case.

**Compliance With Llm Reviewing Policy:**

Affirmed.

**Final Justification:**

This paper still feels to me like it is gluing together two literatures without worrying sufficiently about how they fit. That the "no harm" property precludes any change when defined relative to a trained model, and must be defined relative to a naive/featureless model to correspond naturally to the bargaining literature makes me appreciate it less as a merit relative to the existing fairness literature. I maintain my score.

**Key Questions For Authors:**

1) The authors motivate their solution over a maximin solution by noting that their solution allows for a "no harm" property relative to a default model f_0. But (if I understand correctly) an implication of this is that the author's proposed solution will therefore only differ from the default model if the default model is not already pareto optimal (otherwise any other solution would necessarily harm at least one group). But what baseline model would not already be pareto optimal? Pareto optimality is guaranteed by e.g. standard minimization of average loss, so the "starting point" of most ML problems before fairness constraints are imposed will already be pareto optimal. This seems to me to make the guarantees relative to a default model quite limited.

2) It would be good if something could be said about the leximin optimization problem. These kinds of problems are delicate but have been studied from both an optimization and generalization perspective, e.g. here and in follow ups: https://arxiv.org/abs/2102.08454 --- can techniques from here be applied? This should be discussed.

**Limitations:**

yes

**Strengths And Weaknesses:**

Strengths:
The authors make a connection to the bargaining literature in economics, although this is not entirely novel. For example, Zafar et al. "From Parity to Preference-based Notions of Fairness in Classification" connects fairness in machine learning to the Nash bargaining solution. The present paper is however the first paper I am aware of to connect ML fairness to bargaining solution concepts beyond Nash. Still the authors should cite Zafar et al. and other related work in that line.

Weaknesses:
The authors do little other than making this connection and noting the existing axiomatic characterization of Kalai-Smorodinsky. There is no empirical component to the work, and no algorithmic component. They say nothing about the leximin optimization problem which is necessary for m > 2 groups, and is the main non-trivial issue for both optimization guarantees and sample complexity bounds.

---

> ### Author Rebuttal · Authors · 2026-03-31
>
> We thank Reviewer tqDM for the careful reading and helpful questions.
>
> **Q1:Choice and Role of Baseline.** We will add Appendix C to discuss $f_0$ in detail, covering three aspects: (i) interpretable default choices and their connection to explained variance, and (ii) the bargaining interpretation of $f_0$ as a disagreement point. The following summarizes the key points of part (i).
>
> **In most applications, $f_0$ should be chosen as an interpretable baseline that represents the trivial prediction made without access to covariates which, except for degenerate cases, is not Pareto optimal.** In regression, the natural choice is the unconditional mean $f_0(x) = 0$ (after centering); under squared loss, the denominator becomes $R_g(f_0) - R_g(f_g^{\ast}) = \Vert\beta_g\Vert_{\Sigma_g}^2$, the group-specific explained variance, and $\rho_g(f)$ coincides with the ratio of explained variance, admitting the interpretation $\rho_g(f) = R^2_g(f) / R^2_g(f_g^{\ast})$ where $R^2$ denotes the coefficient of determination. Our baseline choice is consistent with the baseline used by Meinshausen \& B\"uhlmann (2015). In binary classification, the marginal class probability $f_0(x) = \pi_0$ serves the same role.
>
> We will also add a bargaining interpretation paragraph (part (ii)) to clarify why a Pareto-optimal predictor is generally unsuitable as the baseline.
>
> >In the bargaining formulation, $f_0$ represents the disagreement point: the outcome groups revert to if negotiation fails. Crucially, $f_0$ is exogenous to the bargaining problem---it is fixed before negotiation begins and is not itself a product of optimization over $\mathcal{F}$. This further motivates the interpretability requirement. A predictor that already embodies a compromise across groups---such as a pooled ERM solution or any Pareto-optimal predictor---is generally unsuitable as $f_0$ for two reasons. First, $R_g(f_0) - R_g(f_g^{\ast})$ loses its interpretability as a measure of available signal, since it conflates the available signal with the effects of a prior modeling choice. Second, a group that is already satisfied with the existing compromise would have no incentive to participate in the bargaining process, undermining the cooperative framing.
>
> We note that this concern may arise from a natural ambiguity in the choice of $f_0$, which we have now clarified in detail above. The disagreement point $f_0$ is the unconditional mean or marginal class probability---not the average-loss minimizer. Such covariate-free predictors are not Pareto optimal in general: since covariates are typically informative for all groups, there exist predictors $f \in \mathcal{F}$ that improve on $f_0$ for every group simultaneously, so the no-harm guarantee has meaningful content.
>
> Even if we use this interpretable baseline, there can be rare cases in which it is Pareto optimal. In this case, the proposed solution coincides with the default model. This is the only risk vector satisfying the no-harm guarantee, which shows that our relative improvement formulation handles this degenerate case consistently.
>
> **Q2:Leximin optimization.** We thank the reviewer for this pointer and will cite this line of work.  Our paper focuses on the formulation and axiomatic characterization of the fairness criterion rather than its algorithmic realization; Theorem 4.3 establishes well-posedness of the leximin refinement, and we leave the algorithmic extension for future work as noted in the Discussion. The referenced techniques are potentially applicable, but adapting them requires additional analysis due to the group-specific normalization by oracle gaps $R_g(f_0)-R_g(f_g^{\ast})$.
>
> **W1:Empirical validation.** Since Reviewer HdaB raises the same empirical concern directly, we refer to our response to Reviewer HdaB (**New real-data experiment**) for the full description of the added ACS study. Briefly, the new analysis shows that differential predictability across groups is common in practice and can materially affect the selected fairness solution.
>
> **Citation.** We thank the reviewer for pointing out Zafar et al., which we will cite. We agree that it is important related work. Their paper is related in spirit in the sense that it advocates for Pareto-improving constraints inspired by the bargaining problem. The main difference is they don't explicitly calculate the Nash bargaining solution, nor do they axiomatically characterize their solution. By contrast, our paper explicitly maps shared-predictor fair learning onto the bargaining framework with a full axiomatic characterization.
>
> **Contributions.** We want to emphasize that the connection between Fair ML and bargaining solutions is not trivial (Theorem 3.4), and axiomatic characterization enables comparison between widely adopted fairness solutions (see reviewer CqUk (**Axioms comparison**) response). We will highlight our contributions in the introduction (see reviewer CqUk (**Contributions**) response).

---

> > ### Author Rebuttal · Reviewer_tqDM · 2026-03-31
> >
> > Thanks for the response.
> >
> > If we must choose a very weak base model (e.g. one without access to any features) for the guarantees to be nontrivial then the no harm property relative to this model seems correspondingly weak.
> >
> > I agree the contributions of this paper are nontrivial.

---

> > > ### Author Response · Authors · 2026-04-01
> > >
> > > We wish to clarify that a strong base model (e.g., a base model that is on the Pareto front as you suggested earlier) is pathological. The bargaining analogue of a base model on the Pareto front is a disagreement point that is on the Pareto front of the bargaining solution set. In such bargaining problems, the players have no incentive to bargain.
> > >
> > > We also wish to clarify that the no-harm property is intended as a very basic guarantee (i.e., most reasonable solution concepts should satisfy this property).

---

### Official Review · Reviewer_CqUk · 2026-02-27

**Soundness:** 3
**Presentation:** 2
**Significance:** 2
**Originality:** 2
**Overall Recommendation:** 4
**Confidence:** 3

**Summary:**

It is proposed to embed the selection of a "fair" predictor that is used simultaneously for potentially diverse subpopulations as a problem of cooperative bargaining theory. It is then shown that many of the known approaches to tackle this problem within ML (GDRO, MMV, MMR) can be understood as well-known solution principles within this embedding. Special attention is paid to the proposed criterion of relative improvement, which translates back to the KS-solution under the embedding in cooperative bargaining theory. The properties of this novel criterion are then studied in detail from an axiomatic perspective, utilizing among new ideas also existing knowledge about the KS-solution. Finally, the paper studies sufficient conditions for finite sample guarantees for the empirical variant of the proposed relative improvement criterion.

**Compliance With Llm Reviewing Policy:**

Affirmed.

**Final Justification:**

After reading the explanations and proposed changes in the authors' rebuttal, I am now more convinced of the paper and raise my score from 3 to 4.

**Key Questions For Authors:**

1. Would it be possible to showcase the results on real-world application example?

2. Why is relative improvement the most appropriate solution concept compared to other bargaining-theoretic criteria (such as MMR, MMV, GDRO), and can the authors clarify whether the comparison via the (modified set of) axioms is fair—for example, by discussing which of these alternative criteria might satisfy more axioms under different or tailored axiomatic requirements?

3. Given that several theoretical results appear to restate known findings in a new vocabulary, can the authors better highlight their novel and original contributions and relate them to more strongly to the motivating problem of fair learning with diverse suppopulations?

**Limitations:**

Potential improvements and avenues for future research are discussed.

**Strengths And Weaknesses:**

Strengths:

1. In principle, I find the idea of embedding the problem of fair learning with diverse subpopulations in a well-researched field such as cooperative bargaining theory, thereby enabling knowledge transfer between the areas, convincing.

2. The embedding that has been carried out not only allows criteria (such as the KS solution) to be applied to learning problems with diverse suppopulations, but also offers the possibility of examining these (and other more ad hoc) criteria axiomatically. This represents a clear merit, as it creates a framework within which the various criteria (including existing more ad hoc ones) can be systematically categorized and evaluated.

3. I think it's good that the paper also discusses finite sample guarantees and doesn't just discuss the properties of the population variant of the proposed criterion. This adds some more practical value to the contribution.


Weaknesses:

1. I think the contribution is not very well motivated. The only motivation for the problem is a relatively generic and brief credit scoring example, which is not revisited later on. The relevance of the problem for (at least some parts of) the machine learning community is hardly explored. Also, I find the linear regression example described only in the caption of Figure 1 a bit hard to follow.

2. It is not really clear to me why relative improvement is a more suitable criterion than the other solutions that can be transferred from bargaining theory. It is mentioned directly after Theorem 4.1 (“no harm”) that this result does not apply to minimax regret, for example, which is a good argument. However, the table with the axioms on page 7 suggests that each of the criteria fulfills different subsets of these desirable properties, with none of them seeming to stand out in particular. In particular, the comparison seems somewhat unfair to me, as it is not clear to me whether the other criteria could also satisfy a larger subset of these by tailoring the axioms to them specifically.

3. Many of the results of the paper seem to be existing results in a new vocabulary. This somewhat limits the originality of some of the results presented in the paper. This does not have to be an issue per se, but in the light that the papers contribution is purely theoretical without any empirical evaluation or demonstration (see 4. below) becomes more severe in my opinion.

4. In my opinion, the paper would benefit greatly from a real-world application example. Page 2 briefly mentions that the framework proposed here could be applied to a wide range of problems (regression, classification,...), but none of this is actually implemented and showcased in a practical example.


Minor comments:

- Equation (2): \cal P is undefined
- The sentence "We do not make..." in lines 089-091 seems to need improvement.
- line 091: \cal F is undefined
-lines 186 and 203-204 seem to contradict each other: first line says KS-solution is Pareto optimal, other lines say it might be Pareto dominated. (I think the 2-player condition as to be stated earlier)

---

> ### Author Rebuttal · Authors · 2026-03-31
>
> We thank the reviewer CqUk for these thoughtful and constructive questions. They helped us clarify and further develop several parts of the paper.
>
> **W1/Q1:Motivation.** We agree that the original motivation was insufficiently concrete. We have replaced the credit scoring example and Figure 1 with a real-data illustration based on the 2018 ACS Income data; see our response to Reviewer HdaB (**New real-data experiment**) for the revised introduction text. The new Figure 1 will display per-group risk and relative improvement curves for the North Dakota example, making the construction directly grounded in real data.
>
> **W4/Q1:New real-data experiment.** Since Reviewer HdaB raises the same empirical concern directly, we refer to our response to Reviewer HdaB (**New real-data experiment**) for the full description of the added ACS study. Briefly, the new analysis shows that differential predictability across groups is common in practice and can materially affect the selected fairness solution.
>
> **W2/Q2.Axioms comparison.** We want to clarify that Table 1 should be read as a decision guide, not as a claim that RI universally dominates all alternatives. We will add Remark 4.5 to Section 4.4:
>
> >Remark 4.5.(Axiom-based criterion selection)
> No single fairness criterion dominates all others across every setting---Table 1 reflects this by showing that each solution is uniquely characterized by a distinct subset of axioms. Rather than asserting that the relative improvement is universally preferable, we view the axiomatic framework as a *decision guide*: a practitioner should first identify which axioms are most relevant to their context, and the solution is then uniquely determined. For example, when groups differ substantially in inherent predictability, scale invariance is essential---making relative improvement the natural choice. Conversely, a practitioner who prioritizes independence of irrelevant alternatives over individual monotonicity is led to the Nash solution instead.
>
> We note that each row of Table 1 summarizes a characterization theorem: the listed axioms are precisely those that uniquely determine that solution, so the comparison does not favor RI by design.
>
> **W3/Q3:Contributions.** We respectfully disagree with the statement that our contributions "appear to restate known findings in a new vocabulary." In fact, we propose: (i) a new fairness criterion (i.e., Equation (3)), (ii) a novel connection of fair ML with bargaining solutions (i.e., Theorem 3.4 and 4.3), (iii) a unified axiomatic comparison between widely adopted robust group optimization methods, and (iv) provide learning guarantees for the proposed setup (i.e., Theorem 4.1 and 5.4). These are all novel contributions that strongly connect with the fair ML literature. By contrast, only Propositions 4.2 and 4.4 are adaptations of the KS and Imai characterizations, and we do not claim them as entirely novel.
>
> To make our contributions more transparent, we will add the following bullet points to the introduction. Our contributions are:
>
> >- **Relative improvement as a fairness criterion.** When groups differ in inherent predictability, absolute criteria are not comparable across groups. We propose relative improvement $\rho_g(f)$ and show that maximizing its worst case is exactly the Kalai--Smorodinsky bargaining solution, providing an axiomatic justification through scale invariance.
> >- **A unified bargaining framework.** We show that existing robust optimization methods correspond to classical bargaining solutions under a common mapping. Theorem 3.4. establishes the geometric well-posedness needed to import bargaining theory into fair learning.
> >- **Axiomatic characterization as a decision guide.** Each criterion is uniquely characterized by a distinct subset of axioms (Table 1), making their normative commitments explicit and directly comparable within a single framework.
> >- **Learning-specific guarantees.** We prove that relative improvement satisfies individual rationality (no group is harmed relative to baseline), unlike minimax regret, and establish an $O(1/\sqrt{n})$ finite-sample convergence guarantee.
>
> We will also add the following sentence to Section 3.4 to clarify the broader role of Theorem 3.4, which is not a restatement of a classical bargaining result:  classical bargaining theory takes compactness, convexity, and Pareto well-posedness of the feasible set as primitive assumptions, *whereas our contribution is to derive these properties from assumptions on the function class and the loss*:
>
> > Beyond well-posedness, this result provides a general bridge between cooperative bargaining theory and fair learning: under Assumption 3.2 or 3.3, any bargaining solution that selects a Pareto-optimal outcome can be imported into the fair-learning setting without case-by-case verification of geometric well-posedness.
>
> **Minor comments.** We will correct the presentation issues noted in the minor comments.

---

> > ### Author Rebuttal · Reviewer_CqUk · 2026-04-02
> >
> > I thank the authors for their detailed response.
> >
> > My statement that the main contribution was to translate old results into new terminology was somewhat exaggerated; on that point, I agree with the authors.
> >
> > Also, I think the proposed changes are promising. However, as in sum these would be major changes to the manuscript, I still have some concerns about whether they are fully implementable in the final version of the paper.
> >
> > I nevertheless raise my score from 3 to 4.

---

> > > ### Author Response · Authors · 2026-04-02
> > >
> > > We understand this concern and are happy to clarify that these changes are not merely planned: they are already incorporated in our current revised manuscript draft and will be reflected in the final version. As described in our earlier responses regarding the motivation, empirical validation, axioms comparison, and presentation of our contributions, the revised draft now includes an updated Introduction and Figure 1, a concise statement of our contributions at the end of the Introduction, a clarifying paragraph after Theorem 3.4, Remark 4.5, and a new empirical section with additional details in the appendix.
> > >
> > > Thank you again for your thoughtful feedback.

---

### Official Review · Reviewer_HdaB · 2026-03-09

**Soundness:** 3
**Presentation:** 3
**Significance:** 3
**Originality:** 3
**Overall Recommendation:** 4
**Confidence:** 2

**Summary:**

The authors aim to reinterpret group fairness as a bargaining problem where fairness is defined as the relative improvement of achievable risk reduction. They show this corresponds to exactly the Kalai-Smorodinsky (KS) bargaining solution, establish axiomatic properties, situate existing methods including group DRO, MMV, MMR, within the same framework and prove additional fairness guarantee.

**Compliance With Llm Reviewing Policy:**

Affirmed.

**Key Questions For Authors:**

As listed above:

1. The paper suggests f0 is the natural default, such as the mean response in regression or the majority class in classification. Does the choice of different baselines affect, or, to what extent does it affect, the disagreement point, potential improvement denominators and subsequent results?

2. Comment on the lack of empirical validation.

**Limitations:**

yes

**Strengths And Weaknesses:**

Soundness: The theoretic framing of this work is generally solid. One specific question is around the baseline choice f0. The paper suggests f0 is the natural default, such as the mean response in regression or the majority class in classification. Does the choice of different baselines affect, or, to what extent does it affect, the disagreement point, potential improvement denominators and subsequent results?

Presentation: The framework of this paper is generally well-paced and logical.

Significance: I understand the authors have wanted to frame this paper as a purely theoretical one, but from my point of veiw it is still a significant omission to not conduct any empirical validation as a fairness paper. Without the real data experiments, it is less clear whether the proposed criterion makes a meaningful practical difference in the real-world fairness benchmarks.

Originality: the identification of relative improvement with the KS solution is to my knowledge novel in the fair ML literature. And it is useful to have a unified bargaining framework that connects the new fairness concept with group DRO, MMV and MMR.

---

> ### Author Rebuttal · Authors · 2026-03-31
>
> We thank Reviewer HdaB for the positive overall assessment.
>
> **Q1:Choice and Role of Baseline.** The dependence of the MMRI solution on $f_0$ is by design. Changing $f_0$ shifts both the disagreement point $d_g = -R_g(f_0)$ and the potential-improvement denominator $R_g(f_0) - R_g(f_g^*)$, so the solution may change accordingly. We view this as appropriate: relative improvement measures how much of the available signal a model captures, and "available" is defined relative to $f_0$. Since Reviewer FjWq raises the role of the baseline predictor directly, we refer to our response to Reviewer FjWq (**Choice and Role of Baseline**) for the fuller discussion; briefly, $f_0$ is part of the problem specification rather than an arbitrary reference point, and changing $f_0$ defines a different bargaining problem rather than a robustness variant of the same one.
>
> **Q2:New real-data experiment.** We agree that empirical validation is important. To address this concern, we will replace the synthetic introduction example with a real-data illustration. The revised introduction will read (replacing the third and fifth paragraphs of the Introduction):
>
> > However, these worst-case criteria rely on absolute comparisons of loss or regret across groups, implicitly assuming that a unit of reduction in loss is equally meaningful for all groups. To illustrate, consider predicting income for white and non-white workers in North Dakota using age as the feature in the 2018 American Community Survey (Ding et al., 2021): non-white workers have roughly ten times more potential improvement---the gap between the unconditional mean predictor and group-optimal performance---than white workers. Ensuring equal absolute reductions would extract nearly all available signal from the white group while poorly serving the non-white group.
>
> > Figure 1 revisits the North Dakota example using this definition. Minimax regret yields (-74%, 82%): it allocates nearly all model capacity to the non-white group and leaves the white group worse than baseline. Maximin relative improvement equalizes proportional gains at (57%, 57%). This demonstrates that absolute worst-case criteria can yield models extracting vastly unequal fractions of available signal, whereas relative improvement avoids this failure mode.
>
> We will also add a new empirical section (Section 6 and Appendix G) that asks whether differential predictability across groups---the key motivation for relative improvement---arises systematically in practice rather than only in synthetic examples. The following results will be added in Section 6.
>
> > We use the 2018 American Community Survey (ACS) accessed via folktables (Ding et al., 2021), predicting log-transformed personal income for full-time employees across all 50 U.S. states. We consider two binary group partitions (sex, race) and four features (age, education, marital status, householder status), yielding 400 configurations in total. For each, we fit group-reweighted linear models by sweeping $\lambda \in [0,1]$ over 10,001 values to trace the full Pareto frontier.
> >
> > **Differential predictability across groups is common in practice.** For each configuration, we compute the estimate of oracle gap ratio $\hat{r} = (\hat{R}_0(f_0) - \hat{R}_0(\hat{f}_0^{\ast})) / (\hat{R}_1(f_0) - \hat{R}_1(\hat{f}_1^{\ast}))$. Across all 400 configurations, $\hat{r}$ ranges from 0.04 to 3.48, with a median of 0.80. Only 27% of configurations are near-symmetric ($0.8 \leq \hat{r} \leq 1.25$); the remaining 73% exhibit moderate-to-extreme asymmetry, with 26% exceeding a factor of two ($\hat{r} < 0.5$ or $\hat{r} > 2.0$). Asymmetry is especially pronounced under the race partition, where features such as marital status and householder status yield extreme ratios in over 50% and 86% of states, respectively.
> >
> > **Consequences for fairness criteria.** In the North Dakota race partition using age as the single feature ($\hat{r} \approx 0.10$), age is roughly ten times more predictive for non-white workers than for white workers. At the MMR solution, the white group attains -74% relative improvement---that is, MMR actively harms the white group relative to baseline. MMRI equalizes relative improvement at 57% for both groups, avoiding this failure by construction (Theorem 4.1). Full results across states, features, and the four-feature model will be provided in Appendix G.
>
> | Method | $\hat{R}_0$ (White) | $\hat{R}_1$ (NW) | $\hat{\rho}_0$ (White) | $\hat{\rho}_1$ (NW) |
> |--------|--------------|------------|-----------------|--------------|
> | ERM | 0.737 | 0.670 | 95% | 35% |
> | Group DRO | 0.736 | 0.680 | 100% | 21% |
> | MMR | 0.749 | 0.634 | $-$74% | 82% |
> | **MMRI** | **0.739** | **0.653** | **57%** | **57%** |
> | MMV | 0.736 | 0.680 | 100% | 21% |
> | Nash | 0.738 | 0.661 | 80% | 47% |
>
> *North Dakota, race partition, age (AGEP), gap ratio $\approx 0.10$.*

---

> > ### Author Rebuttal · Reviewer_HdaB · 2026-04-03
> >
> > Thank you for the detailed response.

---

### Official Review · Reviewer_FjWq · 2026-03-13

**Soundness:** 3
**Presentation:** 3
**Significance:** 2
**Originality:** 3
**Overall Recommendation:** 4
**Confidence:** 3

**Summary:**

The paper studies fairness through a game-theoretic / bargaining perspective. It proposes a maximin relative-improvement criterion, where each group’s gain is normalized relative to its own attainable improvement from a baseline predictor. The goal is to select a shared model that allocates improvement across groups more evenly in relative terms, and the paper positions this criterion within a broader bargaining framework for group risk trade-offs.

**Compliance With Llm Reviewing Policy:**

Affirmed.

**Final Justification:**

See the review and rebuttal acknowledgement.

**Key Questions For Authors:**

N/A

**Limitations:**

Yes

**Strengths And Weaknesses:**

Strengths:

1. The paper is easy to follow and clearly written.

2. The work provides a new perspective on multi-objective learning or fair learning through the proposed concept of relative improvement, by taking each group's attainable advantage or gain beyond a baseline into account in the fairness adjustment.

3. Another strength is the theoretical analysis. In particular, the paper provides structural results on the geometry of the feasible group-risk set, including compactness and Pareto properties, together with useful uniqueness results under suitable assumptions. These are conceptually valuable for connecting the learning problem to the bargaining formulation.

Weaknesses:

1. The contribution mainly lies in proposing relative improvement as a fairness criterion. Therefore, the paper would benefit from a more detailed discussion of this criterion itself, including its advantages, limitations, and conceptual comparison with other fairness criteria or definitions.

2. The relative-improvement criterion depends critically on the choice of baseline. The paper mentions natural defaults such as the mean response in regression or the majority class in classification, but it remains unclear how practical or robust these choices are. Since the fairness notion is defined relative to this baseline, the paper should discuss more carefully how sensitive the resulting solution is to the choice of $f_0$.

3. The proposed fairness notion focuses on improving the relative gains of the worst-benefited groups without accounting for group cardinality. As a result, it may sacrifice substantial aggregate performance on a larger group in order to balance normalized gains. This may be a reasonable bargaining principle, but the paper should justify more clearly why it should be preferred over standard fairness notions based on absolute error or conditional error parity.

---

> ### Author Rebuttal · Authors · 2026-03-31
>
> We thank Reviewer FjWq for the positive overall assessment and the three focused concerns.
>
> **W1:Advantages and limitations of relative improvement.** We summarize the key advantages and limitations of RI, with pointers to where each will be developed in the revised manuscript:
>
> *Advantages:*
> - **Asymmetric potential improvement across groups.** When groups differ in inherent predictability, absolute criteria allocate disproportionate model capacity to the group with the larger oracle gap (Introduction). RI normalizes by each group's potential improvement; by scale invariance (Proposition 4.2), the solution is unaffected by differences in absolute predictability levels across groups. The new empirical section (Section 6) will provide concrete evidence that such asymmetry is common in practice and materially affects the selected fairness solution (see reviewer HdaB (**New real-data experiment**) response).
> - **No-harm guarantee.** When $f_0 \in \mathcal{F}$, no group is left worse off than the baseline, a property minimax regret violates (Theorem 4.1).
> - **Finite-sample guarantee.** The empirical estimator achieves $O(1/\sqrt{n})$ convergence under standard regularity conditions (Theorem 5.4).
>
> *Limitations:*
> - **Baseline improvement requirement.** RI requires $R_g(f_0) > R_g(f_g^{\ast})$ for all groups, i.e., the baseline must be strictly suboptimal for every group. This is satisfied in usual practical settings where the model class offers meaningful improvement over the null model, but may fail in degenerate cases.
>
> Beyond proposing relative improvement as a new fairness criterion, our bargaining framework enables a principled *conceptual comparison* among criteria. We will add Remark 4.5 to Section 4.4, which makes Table 1 directly usable as a decision guide: each criterion is uniquely characterized by a distinct subset of axioms, so a practitioner can identify the appropriate criterion by first identifying which axioms are most relevant to their context.
>
> **W2:Choice and Role of Baseline.** The solution depends on $f_0$ by design. In most applications, however, there is a natural and interpretable default choice, which substantially reduces ambiguity in practice. Changing $f_0$ defines a different bargaining problem rather than a robustness variant of the same one. We acknowledge that this might not be well detailed in the present version of the manuscript and we will add Appendix C explaining the choice and role of $f_0$ in detail, covering two aspects: (1) interpretable default choices and their connection to explained variance and (2) sensitivity of the solution to the baseline choice:
>
> 1. In most applications, $f_0$ should be chosen as an interpretable baseline that represents the trivial prediction made without access to covariates, e.g., as in Meinshausen \& B\"uhlmann (2015). In regression, the natural choice is the unconditional mean $f_0(x) = 0$ (after centering); under squared loss, the denominator becomes $R_g(f_0) - R_g(f_g^{\ast}) = \Vert\beta_g\Vert_{\Sigma_g}^2$, the group-specific explained variance, and $\rho_g(f)$ coincides with the ratio of explained variance, admitting the interpretation $\rho_g(f) = R^2_g(f) / R^2_g(f_g^{\ast})$ where $R^2$ denotes the coefficient of determination. Our baseline choice is consistent with the baseline used by Meinshausen \& B\"uhlmann (2015). In binary classification, the marginal class probability $f_0(x) = \pi_0$ serves the same role.
>
> 2. Since $\rho_g(f)$ is defined relative to $f_0$, the solution $f_{\mathrm{RI}}$ is sensitive to the baseline. This is inherent to the formulation: group-wise improvement is meaningful only relative to a specified starting point. Our use of the general notation $f_0$ reflects this range of applications rather than an arbitrary modeling choice. In practice, $f_0$ is typically determined by domain convention.
>
> **W3:Group cardinality.** We agree that taking group cardinality into account is important in some practical situations, but this reflects a matter of normative preference. There are two broad approaches to fairness in this setting: population-weighted approaches that optimize average performance subject to fairness constraints, and worst-group approaches that optimize protection of the least-served group directly. Our paper studies the latter family. Accordingly, we do not claim that cardinality-insensitive worst-group objectives are universally preferable; rather, we compare relative improvement with other solutions within that paradigm, viewing the two approaches as reflecting different policy priorities rather than one dominating the other. We note the relevant literature on the former paradigm in the Related Work section (Fairness in Machine Learning paragraph).

---

> > ### Author Rebuttal · Reviewer_FjWq · 2026-04-04
> >
> > Thanks for the explanation.

---

### Decision · Program_Chairs · 2026-04-30

**Decision:**

Accept (regular)

**Comment:**

The back and forth on this paper has reached a conclusion that the work is interesting and well developed, but might lack the degree of significance the authors imply it has, especially in the light of the weakness of some of the formal claims. There were concerns about a lack of empirical work (that the authors have rectified) and some questions about whether the maximin criteria was undertheorized as a notion of fairness (which in my view is not a dealbreaker - the notion the authors use is quite natural).

The issue of significance is to me what brings this paper to the borderline.